# Single human B cell-derived monoclonal anti-*Candida* antibodies enhance phagocytosis and protect against disseminated candidiasis

Fiona M. Rudkin[1], Ingrida Raziunaite[1,6], Hillary Workman[2], Sosthene Essono[2,7], Rodrigo Belmonte[1,8], Donna M. MacCallum [1], Elizabeth M. Johnson[3], Lisete M. Silva[4], Angelina S. Palma[5], Ten Feizi[4], Allan Jensen[2,9], Lars P. Erwig[1,10] & Neil A.R. Gow[1,11]

The high global burden of over one million annual lethal fungal infections reflects a lack of protective vaccines, late diagnosis and inadequate chemotherapy. Here, we have generated a unique set of fully human anti-*Candida* monoclonal antibodies (mAbs) with diagnostic and therapeutic potential by expressing recombinant antibodies from genes cloned from the B cells of patients suffering from candidiasis. Single class switched memory B cells isolated from donors serum-positive for anti-*Candida* IgG were differentiated in vitro and screened against recombinant *Candida albicans* Hyr1 cell wall protein and whole fungal cell wall preparations. Antibody genes from *Candida*-reactive B cell cultures were cloned and expressed in Expi293F human embryonic kidney cells to generate a panel of human recombinant anti-*Candida* mAbs that demonstrate morphology-specific, high avidity binding to the cell wall. The species-specific and pan-*Candida* mAbs generated through this technology display favourable properties for diagnostics, strong opsono-phagocytic activity of macrophages in vitro, and protection in a murine model of disseminated candidiasis.

[1] Medical Research Council Centre for Medical Mycology at the University of Aberdeen, Aberdeen AB25 2ZD, UK. [2] Global Biotherapeutic Technologies, Pfizer Inc, Cambridge Kendall Square, Cambridge, MA 02139, USA. [3] National Infection Service, PHE South West Laboratory, Science Quarter, Southmead Hospital, Bristol BS10 5NB, UK. [4] Glycosciences Laboratory, Department of Medicine, Imperial College London, Du Cane Road W12 0NN, UK. [5] UCIBIO-REQUIMTE, Department of Chemistry, Faculty of Science and Technology, NOVA University of Lisbon, Lisbon 1099-085, Portugal. [6] Present address: Division of Infection and Immunity, The Roslin Institute and Royal (Dick) School of Veterinary Studies, University of Edinburgh, Edinburgh EH25 9RG, UK. [7] Present address: HiFiBiO, 325 Vassar Street, Cambridge, MA 02139, USA. [8] Present address: MSD Animal Health Innovation AS, Thormøhlensgate 55, N-5006 Bergen, Norway. [9] Present address: H. Lundbeck, Ottiliavej 9, 2500 Valby, Denmark. [10] Present address: Galvani Bioelectronics, 980 Great West Road, Brentford TW8 9GS, UK. [11] Present address: School of Biosciences, University of Exeter, Geoffrey Pope Building, Exeter EX4 4QD, UK. Correspondence and requests for materials should be addressed to N.A.R.G. (email: n.gow@exeter.ac.uk)

Fungi cause approximately 1.5 million lethal infections each year—as many as tuberculosis or HIV, and more than malaria or breast or prostate cancer[1]. Of these fungal diseases, Candida species collectively account for the majority of serious fungal infections and represent the fourth leading cause of healthcare-associated infections in the United States[1,2]. Candida albicans is the most commonly isolated species and represents the most prevalent fungal opportunistic pathogen worldwide[3]. Impairment of host immunity, due to trauma, pharmacological or surgical intervention, or alteration in the natural microbiota, determines the frequency and severity of disease[4]. Late diagnosis of invasive candidiasis using 'gold standard' blood culture methodologies along with limitations in the versatility, accuracy and widespread availability of inexpensive and rapid diagnostic tests contribute to the poor prognosis and high mortality rates associated with septicaemia and invasive fungal disease[5–7]. To make inroads into these high disease burdens and mortality figures, better diagnostics, antifungal drugs, immunotherapies and fungal vaccines are urgently required.

Pooled immunoglobulin from serum was one of the first widely available treatments for microbial infections. For example, hyperimmune human serum immunoglobulin has been used to treat a number of infections including cytomegalovirus, hepatitis A and B virus rabies and measles[8–10]. In recent years, monoclonal antibodies (mAbs) have become some of the world's bestselling drugs, with global sales forecast to reach approximately $125 billion by 2020[11]. To date, the majority of these mAbs have been licensed for the treatment of cancer and autoimmune diseases[12,13], but the revolution in applied mAb research has yet to be focussed on mycotic infections. There is currently only one mAb approved for the treatment of an infectious disease (Synagis; respiratory syncytial virus)[14]. Advances have been made in recent years to generate mAbs to viral and bacterial targets and antibody–antibiotic conjugates have also been explored as novel therapeutics against intracellular bacterial pathogens[15–18]. Protective mAbs for clinically relevant fungi have now been reported but these are almost exclusively murine in origin, and generated via hybridoma technology[10,19–24]. Fully human antibodies would represent highly valuable reagents to explore future immunotherapies targeting medical mycoses.

Increased mAb research in the field of mycotic disease has also led to progress in mAb-based diagnostics including the Aspergillus-specific mAb JF5 for the detection of invasive pulmonary aspergillosis, a Candida albicans germ tube mAb (CAGTA) for deep-seated Candida infection and a new cryptococcal antigen dipstick test[25–27]. Assays detecting the pan-fungal marker β-glucan have been a valuable addition to the armamentarium, but for Candida-specific diagnosis, the application of MALDI-TOF MS (matrix-assisted laser desorption ionisation–time-of-flight mass spectrometry) and the introduction of the T2Candida panel test utilising miniaturised magnetic resonance technology to identify clinically relevant species of Candida have been important[28,29]. However, inexpensive, sensitive and specific point-of-care diagnostics that can accurately detect the major human fungal pathogens are urgently required to inform therapeutic strategies.

There are currently no vaccines for the prevention of fungal infection in the clinic, although experimental vaccines based on fungal cell wall targets are in development[30–32]. NDV-3, a vaccine based on a recombinant fragment of the Als3 cell wall adhesin, has now completed Phase II clinical trials where it demonstrated safety and a reduction in the frequency of symptomatic episodes in women suffering from recurrent vulvovaginal candidiasis[33–36]. This vaccine also demonstrates cross-kingdom protection against Staphylococcus aureus due to structural homology of Als3 with surface adhesin/invasin molecules of S. aureus[37]. A pre-clinical Candida-specific vaccine based on the recombinant N-terminal fragment of C. albicans Hyr1 protein demonstrated efficacy in a murine model of disseminated candidiasis, and more recently cross-kingdom protection against the bacterial pathogen Acinetobacter baumannii through structural homology to cell A. baumannii surface proteins[38–40]. These experimental vaccines are based on neutralising and/or protective antibodies that may be deployed in prophylactic or pre-emptive therapies.

Methods and approaches for the production of mAbs for diagnostic and/or therapeutic use have diversified dramatically in recent years. Early mAbs were mainly of murine origin but were immunogenic in the human host[41,42]. Today, the majority of mAbs used clinically are chimeric, humanised or fully human IgG1 mAbs generated through hybridoma cell lines[12]. Combinatorial display technologies using phage or yeast have also been valuable in generating fully human mAbs[43,44] but these often require a period of in vitro affinity maturation and produce mAbs with randomised heavy and light chain pairings. Recently, retention of native VH and VL pairings through direct amplification of individual VH and VL chain domain genes from in vitro expanded single human B cells has led to the generation of fully human mAbs with increased safety, immunotolerance, efficacy and relevance to human disease in areas where current treatments are suboptimal[45–48].

In the present study we address the urgent requirement for improved diagnostics, treatments and vaccines for fungal infection, by generating bespoke recombinant human antibodies from single human B cells. Human antibody encoding variable domain (V) genes targeting Candida epitopes were cloned from single B cells derived from donors who had recovered from mucosal Candida infections to generate a panel of fully human recombinant IgG1 mAbs that display a range of specific binding profiles to pathogenic Candida species. The mAbs demonstrate opsonophagocytic activity in vitro against C. albicans and the emerging multidrug-resistant pathogen Candida auris, and therapeutic efficacy in vivo in a murine model of disseminated candidiasis. They are also effective in a number of diagnostic formats in recognising Candida antigens. This highlights the translational value of this technology for developing fungal diagnostics and therapeutics, and exploring anti-Candida vaccine development.

## Results

**Fully human recombinant anti-Candida mAbs derived from single B cells.** Fully human recombinant mAbs were generated by direct amplification of VH and VL genes from single B cells in which the native antibody heavy and light chain pairing remain intact, thus preserving the original target specificity and disease relevance[45] (Fig. 1). Through screening of in vitro activated class switch memory (CSM) B-cell repertoires, mAbs were isolated that recognised a recombinant C. albicans-specific protein antigen—the hyphal cell wall protein Hyr1[49], or C. albicans whole cell wall preparations. Hyr1 was selected following pre-clinical data that a recombinant N-terminal Hyr1 fragment conferred protection in a murine model of disseminated candidiasis[38,39]. Whole cell wall preparations were used to generate mAbs to a variety of antigens with a range of reactivities and clinical uses.

To enhance the likelihood of isolating Candida-specific antibodies, CSM B cells were isolated from the blood of individuals who had recovered from a superficial Candida infection (mostly vaginitis) within a year of sampling. Donors were selected from a panel of volunteers and the levels of target-specific circulating IgG in the donor plasma was assessed via enzyme-linked immunosorbent assay (ELISA). In the screen, the donors displaying the greatest IgG activity against Hyr1 (donor 23) and C. albicans whole cells (donor 85) (Fig. 2a–c) were used

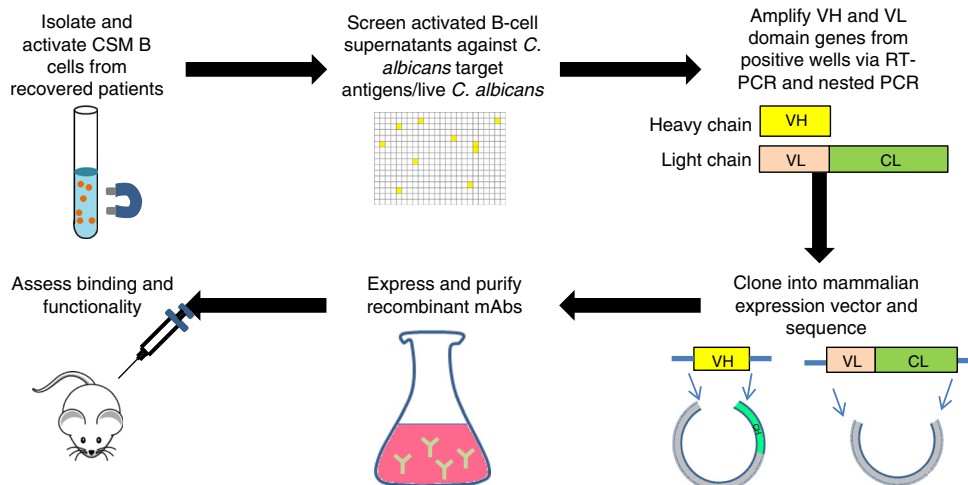

**Fig. 1** Generation of human monoclonal antibodies from single B cells. Class switched memory B cells were isolated from individuals and microcultured in activating media to induce differentiation into plasmblast/plasma cells to promote IgG secretion for screening against target antigens. VH and VL genes from B-cell cultures positive for the target were recovered by RT-PCR and cloned into a mammalian expression vector for expression as full-length human IgG1 followed by standard IgG purification via fast protein liquid chromatography. Following QC, recombinant mAbs were assessed for functional activity in vitro and in vivo. $V_H$: heavy chain variable domain, $V_L$: light chain variable domain, $C_H$: heavy chain constant domain, $C_L$: light chain constant domain

to provide the source of B cells for the generation of mAbs. Approximately 80,000–150,000 CSM B cells were plated out at ≤5 cells/well and activated with a cocktail of cytokines and supplements to promote differentiation to plasmablast or plasma cells associated with secretion of IgG into the culture supernatant[47,50–52]. IgG was detected by ELISA-based high-throughput screening of culture supernatants against target antigens. Typically, 0.05% wells per screen were positive (OD >4× background). Non-specific hits were identified and eliminated by ELISA screening against two unrelated proteins—human serum albumin (HSA) and human embryonic kidney cell (HEK) nuclear antigen. Antigen-positive activated CSM B cells were lysed and used for VH, Vκ-Cκ and Vλ-Cλ gene amplification via reverse transcription polymerase chain reaction (RT-PCR) and nested PCR (representative images shown in Supplementary Figures 3a, b). VH, Vκ-Cκ and Vλ-Cλ genes were subcloned into a mammalian expression vector and corresponding heavy and light chains originating from the same hit well were co-transfected into Expi293F cells for small-scale recombinant whole IgG1 expression. Recombinant mAbs that demonstrated binding to the original target were selected for large-scale expression and then purified via affinity-based fast protein liquid chromatography (FPLC) using a protein A resin before quality control (QC) checking via analytical mass spectrometry, analytical size exclusion chromatography (SEC) and sodium dodecyl sulphate–polyacrylamide gel electrophoresis (SDS-PAGE) gel analysis under non-reducing and reducing conditions (representative images shown in Supplementary Figures 3 c-f respectively). In total, 17 purified recombinant IgG1 mAbs were generated. Five of these bound to purified Hyr1 protein and were split into four clusters defined by their VH CDR3 amino acid sequence. The remaining 12 mAbs bound to C. albicans whole cells and were split into 7 clusters based on their VH CDR3 amino acid sequences (Supplementary Table 6).

**Target-specific binding**. Purified anti-Hyr1 mAbs AB120, AB121, AB122 and AB123 demonstrated strong binding to purified recombinant Hyr1 with half-maximal effective concentration ($EC_{50}$) values between 50 and 100 ng/ml (Fig. 2d). AB124 bound to Hyr1 with lower affinity with an $EC_{50}$ value of 1050 ng/ml. No

binding to HSA and HEK nuclear antigen was observed (Supplementary Figure 4a and 4b). The majority of purified anti-whole cell mAbs bound C. albicans yeast cells with high affinity with $EC_{50}$ values between 3 and 30 ng/ml (Fig. 2e, f). AB134 and AB135, which share high amino acid sequence homology, demonstrated a lower degree of binding with $EC_{50}$ values of 1060 and 220 ng/ml respectively to yeast cells and 684 and 69 ng/ml to hyphal cells (Fig. 2f). Binding avidity for most antibodies screened against yeast and hyphal cell walls was C. albicans morphology dependent (Fig. 2g, h). $EC_{50}$ values were used here to demonstrate the variability in anti-whole cell mAbs binding to C. albicans cell surface antigens. As before, no binding of anti-whole cell mAbs to HSA or HEK nuclear antigen was observed, confirming that their specificity for fungal cells was retained throughout the cloning process (Supplementary Figure 4c-f). Therefore, this methodology generated a panel of mAbs which bound to a variety of fungal-specific cell targets.

**mAbs bind to C. albicans cell surface antigens**. Anti-Hyr1 mAbs were used to stain the C. albicans cell surface by indirect immunostaining. The anti-Hyr1 mAbs bound only to hyphae, and not the parental yeast cells of germ tubes (Fig. 3a). Anti-Hyr1 mAbs did not bind to hyphae of a hyr1Δ/hyr1Δ null mutant and binding was restored in a mutant that was transformed with a single reintegrated copy of HYR1 (Fig. 3a). These antibodies did not stain hyphae of C. dubliniensis—a closely related species to C. albicans that lacks the HYR1 gene.

Immunofluorescent and transmission electron microscopy (TEM)-immunogold staining using anti-whole cell mAbs demonstrated specific and distinct binding patterns to C. albicans cells (Fig. 3b). AB126 and AB131 bound both yeast and hyphae but exhibited very little binding to the mother yeast cells. AB127 bound to mother yeast cells and hyphal apices, but not to the trunk of growing hyphae. AB118, AB119, AB140 and AB135 bound to antigens widely expressed on both C. albicans yeast and hyphae. Therefore, collectively the panel of mAbs detected a range of both morphology-specific and morphology-independent epitopes located in the inner cell wall.

C. albicans cells were treated enzymatically with proteinase K, endoglycosidase H (endo-H), Jack Bean α-mannosidase and

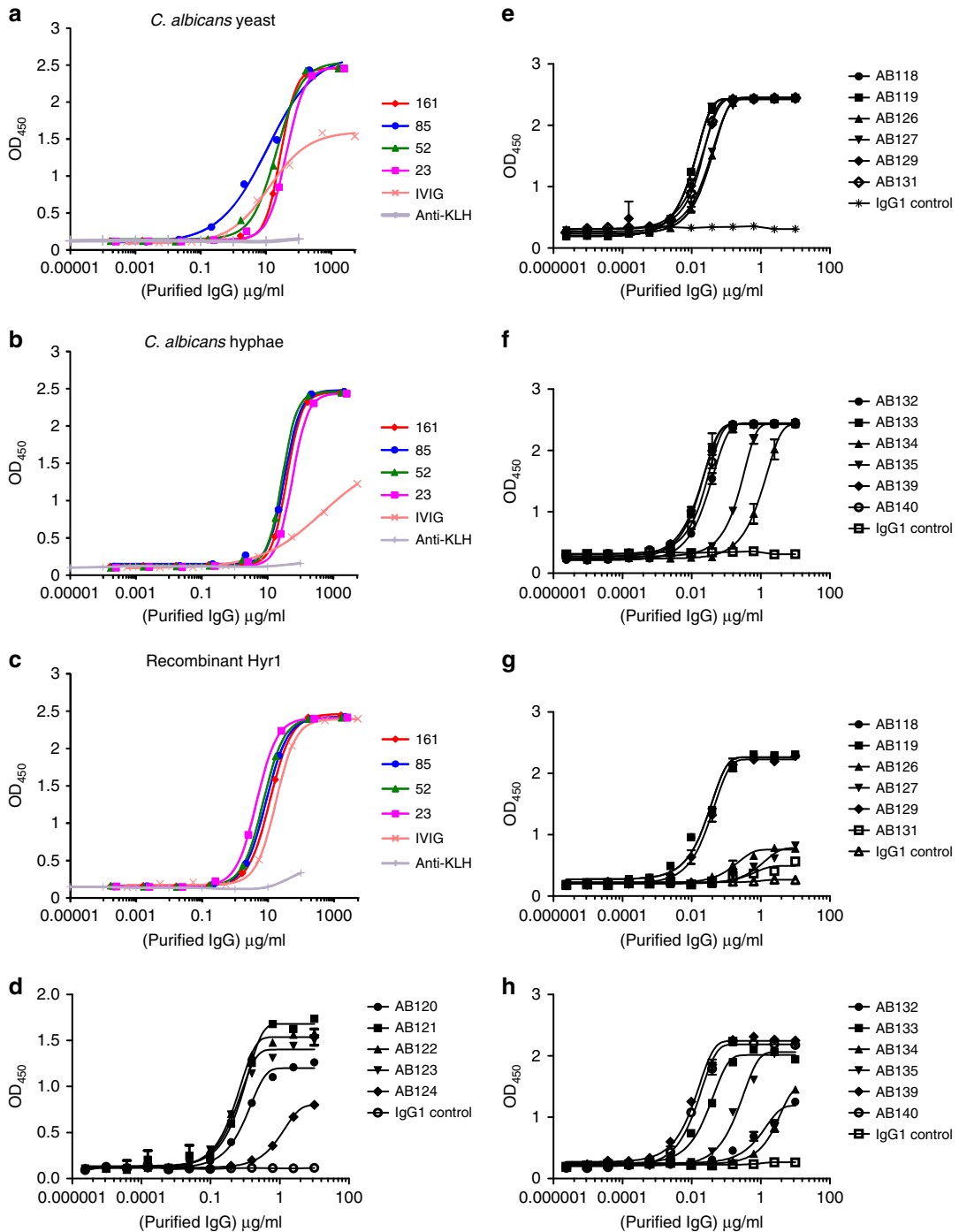

**Fig. 2** Screen of donor circulating IgG and binding of purified anti-Candida mAbs to target antigens. Purified IgG isolated from donor serum was screened against *C. albicans* yeast cell wall extract (**a**), *C. albicans* hyphal cell wall extract (**b**), and recombinant Hyr1 protein fragment (**c**) via ELISA. Values represent means (*n* = 2). **d** Purified anti-Hyr1 mAbs AB120-AB124 binding to purified recombinant Hyr1. **e**, **g** Purified cell wall mAbs AB118-AB131 binding to *C. albicans* yeast and hyphal cells respectively. Binding of the remaining cell wall mAbs AB132-AB140 to *C. albicans* yeast and hyphal cells via ELISA is shown in **f** and **h** respectively. Values represent mean ± SEM (*n* = 2-4)

zymolyase 20T and then assessed for mAb binding. Following endo-H and α-mannosidase treatment, binding of AB119 was decreased, suggesting that the target epitope for this mAb contains a core *N*-mannan structure but not outer chain manno-oligosaccharides (Supplementary Figure 5a). Binding of AB135 was reduced following treatment of cells with zymolyase 20T, suggesting that this mAb targets a *Candida*-specific β-glucan-associated epitope (Supplementary Figure 5b). Proteinase K treatment reduced binding of AB120 (anti-Hyr1) but not

anti-whole cell mAbs to *C. albicans*, confirming that AB120 recognises a protein or proteoglycan epitope (Supplementary Figure 5c). Some anti-whole cell mAbs demonstrated increased fluorescence after enzymatic treatment, suggesting that their epitopes might be located deeper in the cell wall.

**No cross-reactivity to other fungal, bacterial, plant or mammalian glycans**. The mAb with the lowest EC$_{50}$ value from each

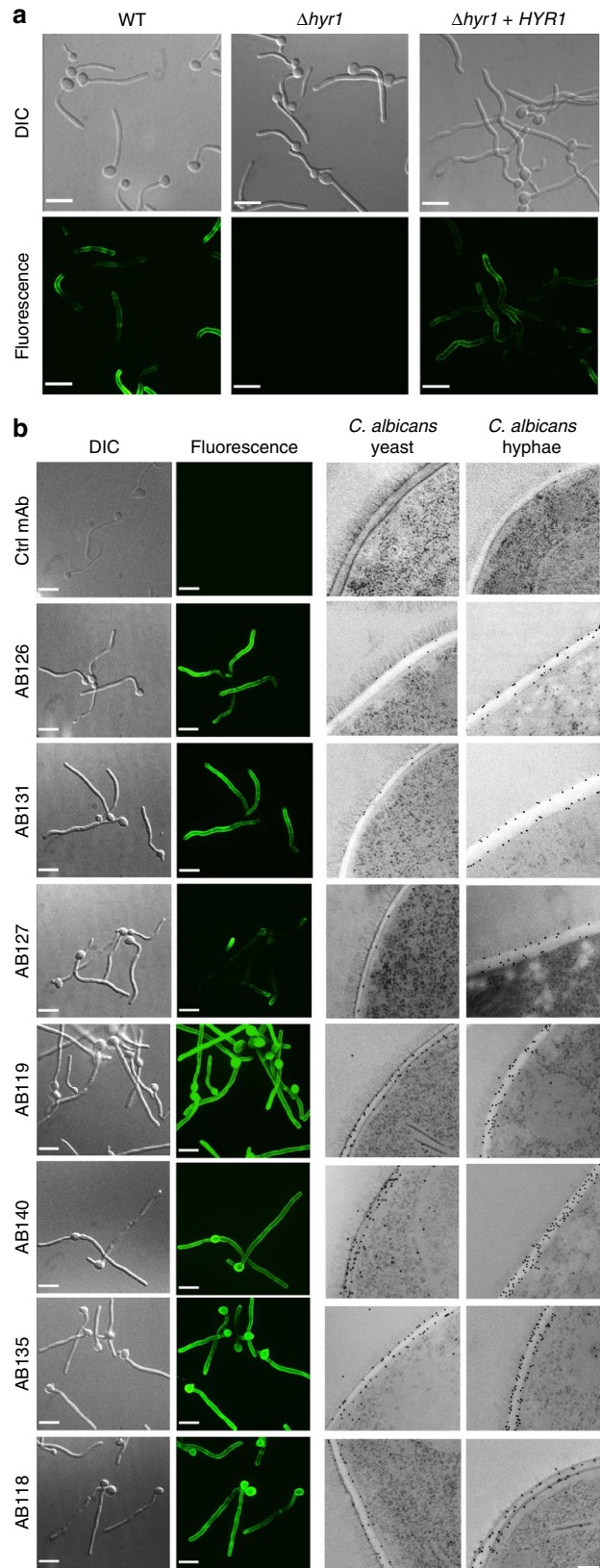

**Fig. 3** Visualising anti-*Candida* mAbs binding to *C. albicans* cells. **a** Representative immunofluorescent images from three separate experiments with anti-Hyr1 mAb AB120 against WT CAI4-CIp10, Hyr1 null mutant and a Hyr1 reintegrant strain. **b** Also shown are immunofluorescent images and corresponding immunogold localisation of anti-whole cell mAbs binding to *C. albicans* yeast and hyphal cell walls. A fluorescently conjugated secondary goat anti-human IgG antibody was used to detect anti-*Candida* mAb binding. Fluorescent images are representative images from three separate experiments. Transmission electron microscopy (TEM) images show representative cells from at least one experiment. Scale bars represent 4 μm on immunofluorescent images and 100 nm on TEM images

accordance with the guidelines as published by the MIRAGE initiative[54]. A MIRAGE Glycan Microarray Document is included as a Supplementary Note in Supplementary Information. All anti-*Candida* mAbs tested (except the protein-specific anti-Hyr1 mAb AB121) demonstrated strong binding to *C. albicans* mannoprotein (ID 13 in *N*-Glycan Array Set 3) and no cross-reactivity to other fungal-type, bacterial or plant oligosaccharides included in the array. The mammalian β-glucan receptor dectin-1, which was included for reference in the analyses, showed the predicted highly specific binding to fungal β1,3-glucans (Fig. 4a and Supplementary Table 7). Additionally, there was negligible or no binding to the mammalian-type *N*-glycans tested (Fig. 4b and Supplementary Table 8). In these analyses the broadly neutralising anti-HIV mAb PGT 128, known to recognise Man9- and Man8-high-mannose *N*-glycans[55], was used as a reference. The results further demonstrated the specificity of the anti-*Candida* mAbs for *Candida*-associated polysaccharide epitope(s).

**mAbs demonstrate distinct binding profiles to other pathogenic fungi.** When assessing anti-whole cell mAb binding to yeast cells of other pathogenic *Candida* species via fluorescence-activated cell sorting (FACS), AB118, AB119, AB135 and AB140 (representing mAbs from four clonal clusters) bound avidly to the most closely related *Candida* species *C. dubliniensis*, *C. tropicalis*, *C. parapsilosis* and *C. lusitaniae* and the emerging pathogen *C. auris*. AB140 also demonstrated specific binding to the more distantly related *C. glabrata*. mAbs which bound to epitopes more highly expressed on *C. albicans* hyphae such as AB126, AB127 and AB131 (representing three clonal clusters) demonstrated overall lower levels of binding to the phylogenetically closest relatives of *C. albicans* (Figs. 5, 6). Interestingly, AB131 also demonstrated low levels of binding to *C. krusei* and *Saccharomyces cerevisiae* (Figs. 5, 6). Immunofluorescent imaging was carried out to assess patterns of mAb binding in hypha-inducing conditions. Here, AB126, AB127 and AB131 demonstrated increased levels of binding to most species and in particular *C. albicans*, commensurate with the predominantly hyphal-associated expression of their target epitopes (Fig. 7b). To assess for pan-fungal binding activity, anti-whole cell mAbs were tested against species from the other three major human fungal pathogens—*Aspergillus fumigatus*, *Cryptococcus neoformans*, *Cryptococcus gattii* and *Pneumocystis carinii*. No binding to any of these species was observed (Fig. 7b).

Commensurate with the *C. albicans*-specific nature of *HYR1*, anti-Hyr1 mAbs bound only to *C. albicans* hyphae and not to *C. albicans* yeast cells or any of the other *Candida* species tested including *C. dubliniensis* which lacks the *HYR1* gene (Figs. 5, 6, 7a). Therefore, the anti-Hyr1 mAbs were *C. albicans* specific and the majority of anti-whole cell mAbs demonstrated a variety of binding patterns to wild-type *C. albicans* and other pathogenic *Candida* species, but not to more distantly related fungal pathogens, indicating that they target a range of *Candida*-

major clonal cluster was chosen as a representative for further screening against polysaccharides and glycoproteins that contain oligosaccharide sequences found in fungal cell walls (Supplementary Table 7) and against mammalian-type *N*-glycans (Supplementary Table 8) in an immobilised glycan microarray format[53]. Glycan microarray studies were conducted in

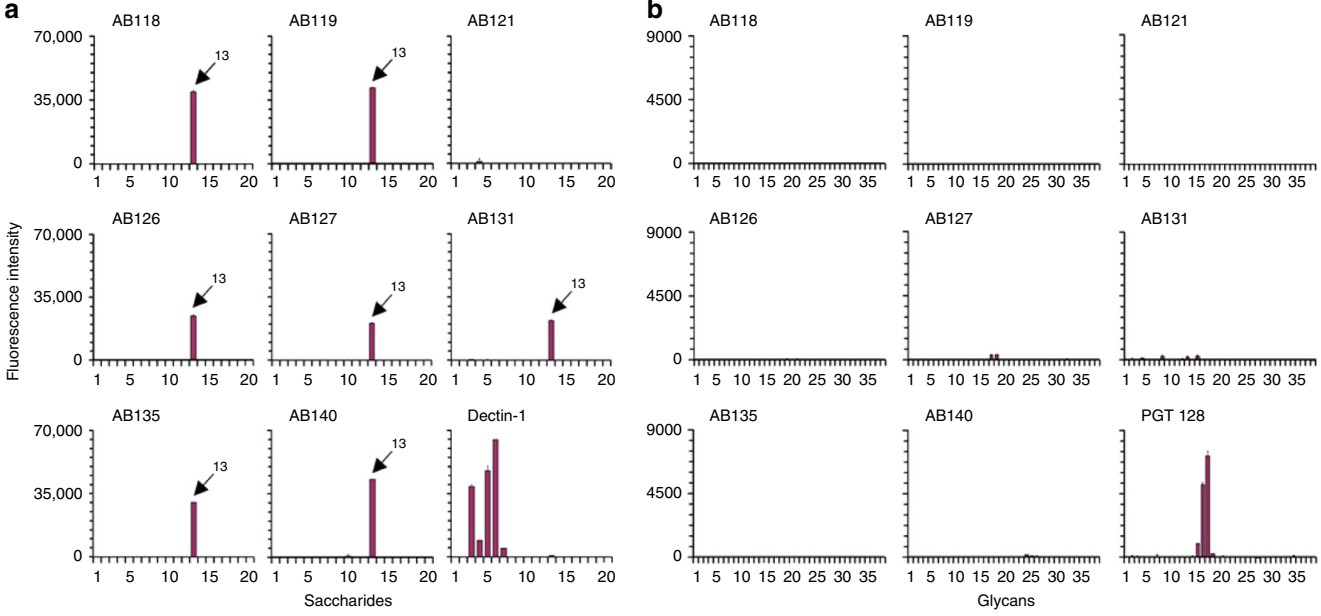

**Fig. 4** Glycan microarray screening analyses of anti-*Candida* mAbs. **a** Binding of the anti-*Candida* mAbs and human Fc-tagged murine dectin-1 to the Fungal, Bacterial and Plant Polysaccharide Array (Supplementary Table 7). The information on the saccharide IDs and predominant oligosaccharide sequences is in Supplementary Table 7. The results are shown as histograms at 0.1 ng/spot (saccharide IDs 1–19) and at 5 fmol/spot (saccharide ID 20). The *C. albicans* mannoprotein (ID 13) is highlighted in the antibody panels. Values represent mean fluorescence intensities ± errors (half of the difference of signal intensities of duplicate spots for each saccharide/glycan probe) (*n* = 2). **b** Binding of the anti-*Candida* mAbs and mAb PGT 128 to the *N*-glycan Array Set 3 (Supplementary Table 8). These arrays are at 5 fmol/spot. Values represent mean fluorescence intensities ± errors (half of the difference of signal intensities of duplicate spots for each glycan probe) (*n* = 2)

specific antigens whose expression levels varies from species to species.

**mAb binding promotes FcγR-dependent phagocytosis by macrophages.** The possible opsonising effect of anti-*Candida* mAbs on phagocytosis was examined by live cell imaging using J774.1 mouse macrophages (European Collection of Cell Cultures (ECACC)). Macrophages were challenged with live, *C. albicans* CAI4-CIp10 which had been pre-incubated with an anti-*Candida* mAb, an IgG1 control mAb or saline for 1 h. *C. albicans* cells that had been pre-incubated with AB119 or AB140 (anti-whole cell mAbs) were cleared significantly more rapidly compared to saline and AB120 or IgG1-treated controls (Fig. 8a). The antibodies therefore strongly influenced macrophage behaviour enabling highly efficient targeting of fungal cells (see Supplementary Movies 1–7). We compared differences in the rate of engulfment of *C. albicans* pre-incubated with selected anti-*Candida* mAbs with isotype controls, defining engulfment as the time taken from the establishment of fungus–phagocyte contact to the time for complete engulfment indicated by loss of fluorescein iso-thiocyanate (FITC) fluorescence (Fig. 8b). Anti-whole cell mAb-bound fungal cells were engulfed significantly faster compared to saline controls and the IgG1 control mAb (Fig. 8c; Supplementary Movies 1–7). *C. albicans* yeast cells pre-incubated with the hypha-specific mAb AB120 were engulfed at a slower rate compared to cells coated with an anti-whole cell mAb but considerably faster than the IgG1 control (Fig. 8c). To examine the effect of AB120 on a hypha-only cell population, AB120 was pre-incubated with filamentous *C. albicans* cells before addition to macrophages. Here, AB120 stimulated significantly faster macrophage engulf-ment of *C. albicans* hyphal cells compared to the saline control (Fig. 8d; Supplementary Movies 6, 7). Binding of the antibodies therefore strongly induced phagocytosis of *C. albicans* by murine J774.1 macrophages. Killing of *C. albicans* cells by J774.1 mac-rophages was unaffected by antibody pre-incubation under these

conditions. Similar observations were obtained using human monocyte-derived macrophages (Supplementary Figure 7). Blocking Fcγ receptors (FcγR) decreased the rate of engulfment of AB140-bound *C. albicans* to a similar level as that of the saline control (Fig. 8e). indicating that the mAb-dependent stimulation of engulfment was FcγR dependent.

Macrophages travelled faster and further towards *C. albicans* yeast cells when pre-incubated with an anti-whole cell mAb (AB140) (Fig. 8f, g) and exhibited more directional movement towards antibody-bound *C. albicans* cells compared to control (Fig. 8h–j). The opsonising effect of anti-whole mAbs on *C. albicans* phagocytosis was also observed with the multidrug-resistant pathogen *C. auris*, where AB119 induced noticeable clearance of fungal cells by 10 min and almost complete clearance of cells by 30 min (Fig. 9; Supplementary movies 8 and 9). This correlated with the high level of binding of AB119 to *C. auris* observed by immunofluorescence imaging and FACS.

**mAb reduces fungal burden in disseminated candidiasis model.** We assessed therapeutic potential of the anti-*Candida* mAbs in two murine models of systemic candidiasis[56]. The first model was designed to test whether mAb efficacy observed in phagocytosis assays in vitro translated into protection in an in vivo model. In this model, *C. albicans* SC5314 yeast cells were pre-incubated for 1 h with either saline, an IgG1 isotype control mAb, AB119 or AB120 before intravenous injection into the mouse lateral tail vein. Disease progression was monitored by weight change and kidney fungal burdens at day 3[56]. When pre-incubated with Hyr1-specific mAb AB120, there was no decrease in fungal burden compared to the saline control or the IgG1 control mAb (Fig. 10a). When the anti-whole cell mAb AB119 was pre-incubated with SC5314 yeast cells, there was also a sig-nificant decrease in kidney fungal burden compared to the saline control (Fig. 10a). Burden results were reflected in mouse body

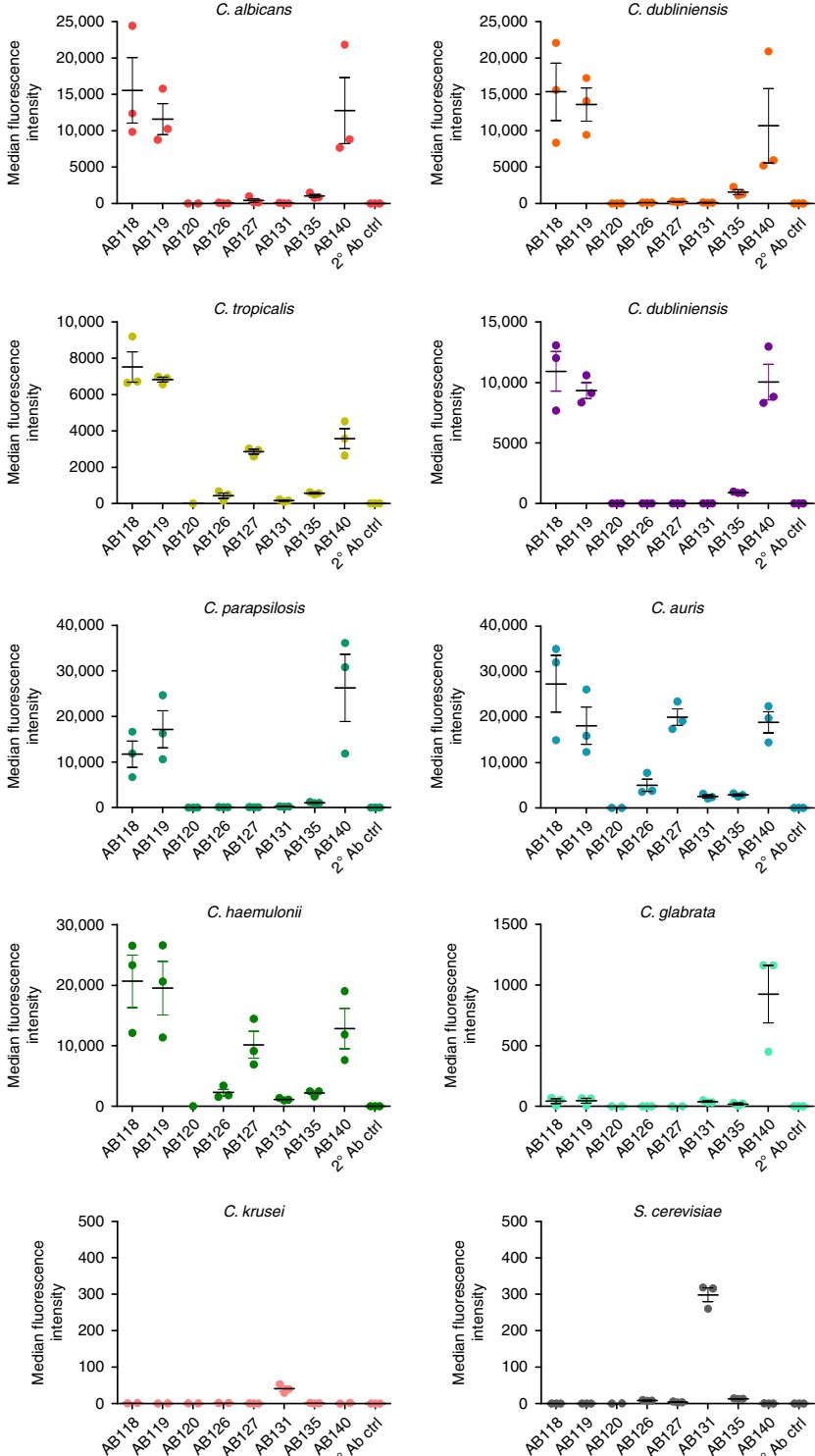

**Fig. 5** Binding of anti-*Candida* mAbs to panel of *Candida* species and *S. cerevisiae*. Binding of anti-Hyr1 mAb and anti-whole cell mAbs (one mAb from each clonal cluster) to yeast cells of different *Candida* species and *S. cerevisiae*. Data were obtained using a BD Fortessa flow cytometer. Dots represent single experiments and horizontal lines represent the mean of median fluorescence intensities ± SEM ($n = 3$)

weights (Supplementary Figure 8a). We also tested the ability of antibody AB119 to protect prophylactically. This study was conducted using a blinded protocol by Evotec (UK) Ltd. An irrelevant IgG1 isotype control mAb or AB119 was administered by intraperitoneal injection 4 h prior to challenge with *C. albicans* SC5314 yeast cells via intravenous injection into the mouse lateral tail vein. Disease progression was measured by kidney fungal burdens and weight change at day 7 following Evotec's standard protocol. Prophylactic treatment with AB119 significantly reduced kidney fungal burdens compared to IgG1 control mAb (Fig. 10b). Burden results were reflected in mouse body weights (Supplementary Figure 8b). Therefore, the anti-whole cell mAb AB119 also demonstrated protection in a prophylactic mouse model of systemic *Candida* infection.

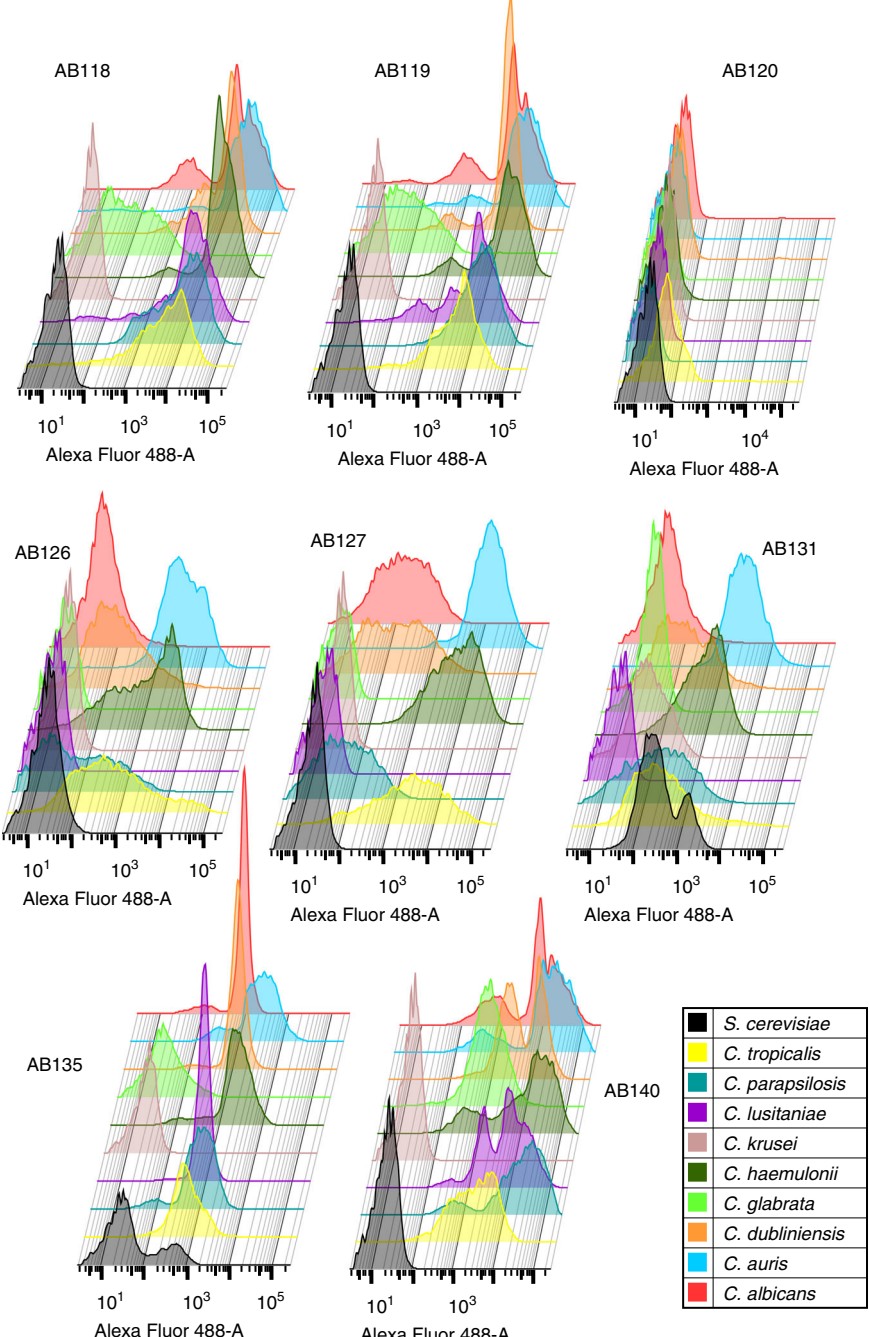

**Fig. 6** Anti-*Candida* mAbs binding profiles to clinically relevant *Candida* species and *S. cerevisiae*. Representative FACS data from three separate experiments of anti-*Candida* mAbs (one mAb from each clonal cluster) binding to different species shown as histograms. Data were obtained using a BD Fortessa flow cytometer

**mAb detection of *C. albicans* antigens in diagnostic assay format**. Selected mAbs were assessed for compatibility in a counterimmunoelectrophoresis diagnostic assay which is commonly employed to diagnose patients with *Candida* infection. AB118, AB119 and AB140 reacted with *C. albicans* yeast antigenic preparations to generate strong precipitin lines (Fig. 10c). When assessed for reactivity to *C. albicans* hyphal antigenic preparations, AB135 gave a positive result through the formation of a clear precipitin line and mAbs AB118, AB119 and AB140 formed weak precipitin lines (Fig. 10d). Together, these data indicate that these mAbs possess potential utility as *Candida* diagnostic agents.

## Discussion

Monoclonal antibodies or oligoclonal mixtures of mAbs have unexplored potential in antifungal therapy and disease management. Antibody-based agents have been identified as an alternative strategy to complement the medical gaps associated with current antifungal diagnostics and treatments[9,57]. We describe the first isolation of fully human anti-*Candida* mAbs using novel single B-cell antibody isolation methodology. These mAbs, derived from single B cells from donors with a history of mucosal *Candida* infection, demonstrated fungal-specific binding profiles to *C. albicans* and other pathogenic *Candida* species and the ability to opsonise fungal cells, thereby enhancing phagocytosis

**a**

| mAb | C. albicans | C. dubliniensis | C. tropicalis | C. parapsilosis | C. lusitaniae | C. glabrata | C. krusei |
|---|---|---|---|---|---|---|---|
| AB120 | | | | | | | |
| AB121 | | | | | | | |
| AB122 | | | | | | | |
| AB123 | | | | | | | |

**b**

| mAb | C. albicans | C. dubliniensis | C. tropicalis | C. parapsilosis | C. lusitaniae | C. auris | C. haemulonii | C. glabrata | C. krusei | A. fumigatus | C. neoformans | C. gattii | P. carinii |
|---|---|---|---|---|---|---|---|---|---|---|---|---|---|
| AB118 | | | | | | | | | | | | | |
| AB119 | | | | | | | | | | | | | |
| AB126 | | | | | | | | | | | | | |
| AB127 | | | | | | | | | | | | | |
| AB131 | | | | | | | | | | | | | |
| AB135 | | | | | | | | | | | | | |
| AB140 | | | | | | | | | | | | | |

High binding No binding

**Fig. 7** Heat-map of anti-*Candida* mAbs binding to *Candida* and other pathogenic fungi. Immunofluorescence microscopy analysis of **a** anti-Hyr1 mAbs (AB120-AB123) and **b** cell wall mAbs (one from each clonal cluster) binding to *C. albicans* and other clinically relevant fungal species under hyphal inducing conditions. Red (high binding) to yellow (no binding)

and exhibiting protection in a murine model of disseminated candidiasis. The antibodies also exhibited potential in the generation of *Candida* diagnostic assays.

We generated 17 fully human recombinant anti-*Candida* mAbs through the direct cloning of VH and VL antibody genes encoded by memory B cells and elicited naturally in vivo in response to a *Candida* infection. This technology circumvents the considerable expense of humanising antibodies isolated from rodents or other sources. Furthermore, the recombinant affinity matured human mAbs generated are less likely to be immunogenic and, importantly, their native antibody heavy and light chain natural pairings remain intact, thus preserving their original epitope specificity as presented during the course of disease and their biochemical properties[48]. IgG1 was selected as the scaffold of choice because it is the most common isotype of therapeutic antibodies in the clinic and is the best characterised in terms of drug development[12,58,59].

Twelve of the mAbs generated bound to *C. albicans* whole cell and 5 bound to recombinant purified Hyr1 protein—a protein that is important for *C. albicans* resistance to phagocytosis and is currently in development as an experimental vaccine[38,39]. Antibodies with species-specific and pan-species activity both have translational potential as diagnostics and therapeutics[6,7]. Therefore, we validated that both highly specific and cross-reactive antibodies can be generated using the same methodology. The anti-Hyr1 mAbs bound only to *C. albicans* hyphae. Most of the anti-whole cell mAbs bound to the closely related pathogens *C. dubliniensis*, *C. tropicalis*, *C. parapsilosis* and *C. auris*. There was less or no binding to the more evolutionarily distant species *C. glabrata* and *C. krusei*. When assessed in a routinely employed diagnostic assay, these mAbs were able to react with *Candida* antigenic preparations. Therefore, the novel technology employed here can be utilised to generate species-specific and/or pan-*Candida* mAbs with considerable potential in antifungal drug discovery and diagnostics.

Many therapeutic mAbs exert their protective effects through opsonising cells for phagocytosis[60]. We observed that *C. albicans* yeast and hyphal cells treated with anti-whole cell mAbs or anti-Hyr1 mAbs were phagocytosed more quickly compared to non-opsonised cells, and that this was FcγR dependent. Furthermore, macrophages migrated further, faster and more directionally towards opsonised *C. albicans* cells and this contributed to the

rapid clearance of fungal cells. Our antibodies could also promote *C. auris* phagocytosis, demonstrating the potential of these mAbs in targeting difficult-to-treat drug-resistant pathogens. Opsono-phagocytic activity was reflected in the observed protective effect of selected antibodies in a 3-day murine model of systemic infection. Humanised or fully human IgG1 mAbs make up the majority of antibody therapeutics used clinically and the IgG1 isotype has been routinely tested pre-clinically in murine models of disease. Human IgG1 also binds to all activating mFcγRs with a similar profile to the most potent IgG isotype in mice, mIgG2a, validating the use of mouse models to assess Fc-mediated effects of hIgG1 mAbs[58]. We utilised a 3-day mouse model of disseminated candidiasis[56] to assess the efficacy of anti-*Candida* mAbs in vivo and observed a significant decrease in kidney fungal burden when *C. albicans* was pre-incubated with an anti-whole cell mAb. We then established that the same anti-whole cell mAb demonstrated protection in a clinically relevant 7-day prophylactic treatment model of disseminated candidiasis.

This study describes the first generation of fully human mAbs to a fungal pathogen cloned from genes from single B cells. These mAbs included those with specific pan-*Candida* and species-specific properties that have utility in the antifungal diagnostic and therapeutic sectors. We used *C. albicans* as the screening target but this technology has broad potential in a wide range of other clinical and biological contexts. The characteristics of the mAbs are suitable for use singly or in multiplex formats to create novel polyvalent diagnostic tests. As therapeutics they could be used to direct the immune system to destroy fungal pathogens in a similar approach to cancer immunotherapy, or by targeting toxic molecules to specific microbial or cellular targets. They also have potential in exploring novel antifungal vaccines through the identification of protective antigens.

## Methods

**Ethics statement**. Human blood samples were taken from donors according to the local guidelines and regulations, approved by the Pfizer Institutional Review Board and/or College Ethics Review Board of the University of Aberdeen (CERB/2012/11/676). Informed consent was obtained from all human participants. Animal experiments were approved by the University of Aberdeen Animal Welfare and Ethical Review Body (AWERB) and were carried out under UK Home Office Project Licence PPL60/4135 and conformed to the European Union Directive 2010/63/EU on the Protection of Animals Used for Scientific Purposes. All animal experiments conducted by Evotec Ltd (UK) were performed under UK Home Office Licence PA67E0BAA, and with local ethical committee clearance.

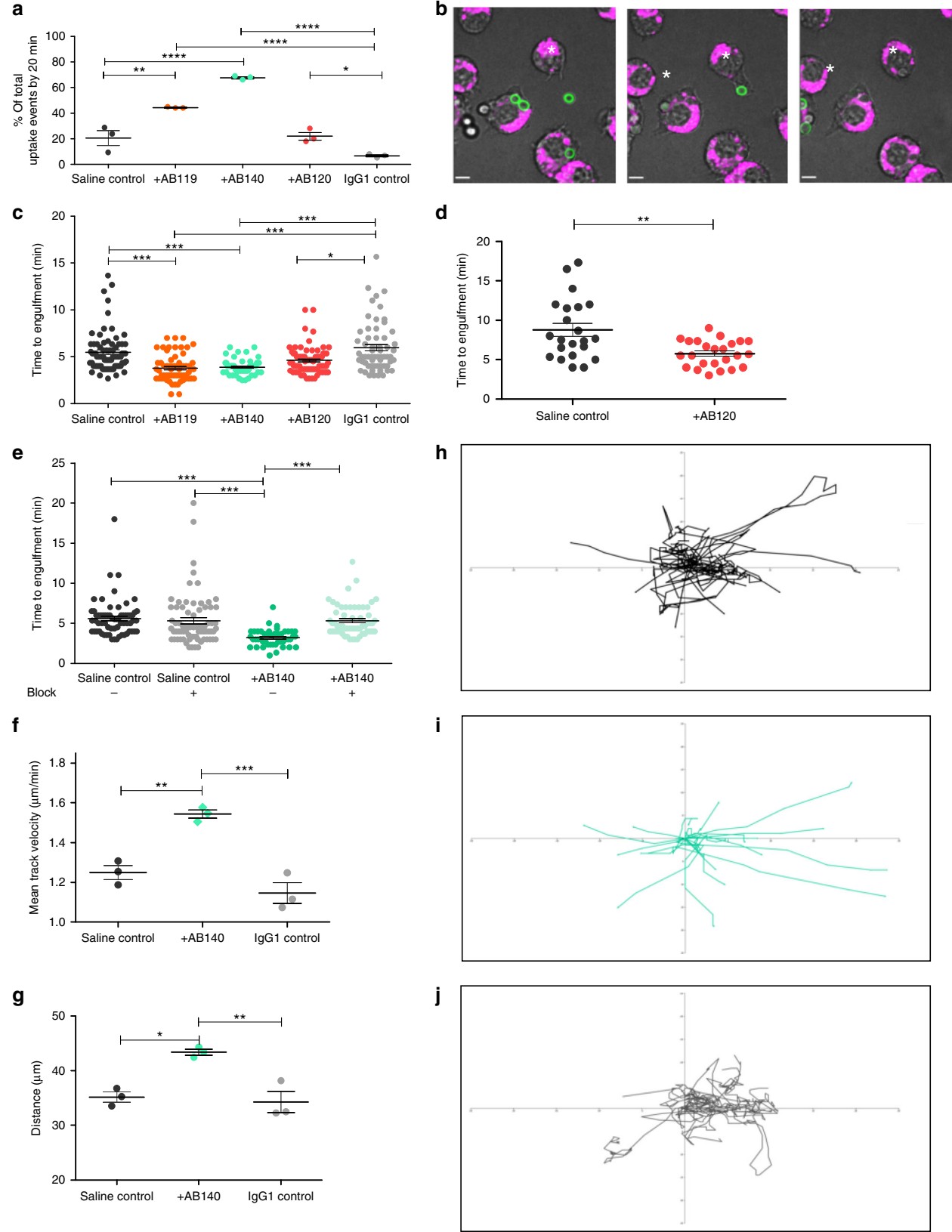

**Fungal strains and growth conditions**. *C. albicans* serotype A strain CAI4+CIp10 (NGY152) was used as the *C. albicans* control and source of antigen and its parent strain CAI4 was used to construct the *hyr1Δ/ hyr1Δ* homozygous null mutant *C. albicans* strain. Other mutants used in phenotypic characterisations were all derived from the CAI4 background[49,61,62]. The *CaHYR1* gene was transformed into *RPS1* locus of the *hyr1Δ/hyr1Δ* null mutant under the regulation of the

Ca*ACT1* promoter to create a *hyr1Δ/hyr1Δ/HYR1* reintegrant heterozygous control strain. Clinical isolates of fungal species used are listed in Supplementary Table 1.

*Candida* strains were obtained from glycerol stocks stored at −80 °C and plated onto YPD plates (2% (w/v) Mycological Peptone (Oxoid, Cambridge, UK), 1% (w/v) yeast extract (Oxoid), 2% (w/v) glucose (Fisher Scientific, Leicestershire, UK) and 2% (w/v) technical agar (Oxoid)). *Candida* strains and *S. cerevisiae* NCPF8313

**Fig. 8** Effect of anti-*Candida* mAbs on macrophage phagocytosis of live *C. albicans* cells. **a** Percentage of uptake events that occurred within the first 20 min of the assay following *C. albicans* pre-incubation with saline, an IgG1 control antibody, an anti-whole cell reactive mAb (AB119 and AB140) or an anti-Hyr1 mAb (AB120). An uptake event was defined as the complete engulfment of a *C. albicans* cell by a macrophage. Dots represent single experiments and horizontal lines represent mean percentage of uptake events ± SEM ($n = 3$). **b** Snapshot images taken from live cell video microscopy capturing the stages of *C. albicans* engulfment by J774.1 macrophages. (left) Macrophage (magenta, *) and *C. albicans* (green) prior to cell–cell contact; (middle) macrophage and yeast cell once contact has been established and (right) *C. albicans* within the phagocyte following ingestion. Scale bar represents 6 µm. **c** Average time taken for a macrophage to engulf a live *C. albicans* cell following pre-incubation with saline, IgG1 control or anti-*Candida* mAb. **d** Average time taken for a macrophage to ingest a filamentous *C. albicans* cell following pre-incubation with saline or AB120. **e** Average time taken for a macrophage to ingest a live *C. albicans* cell following pre-incubation with saline or AB140 in the presence or absence of an FcγR block. **c**, **d**, **e** Dots represent individual uptake events from three experiments; horizontal lines represent mean time taken ± SEM ($n = 3$). **f** Mean velocity of and **g** average distance travelled by macrophages during migration towards *C. albicans* cells following pre-incubation with saline, an IgG1 control mAb or AB140. Dots represent single experiments and horizontal lines represent means ± SEM ($n = 3$). **h**–**j** Tracking diagrams representing macrophage migration towards *C. albicans* cells pre-incubated with saline (**h**), AB140 (**i**) or IgG1 control mAb (**j**). Tracks represent the movement of individual macrophages relative to their starting position, until the first uptake event. Statistical significances were determined using one-way ANOVA with Bonferroni's post hoc test (**a**, **f**, **g**), two-tailed *t*-test with Welch's correction (**d**) or Kruskal–Wallis test with Dunn's multiple comparison test (**c**, **e**); *$P < 0.05$, **$P < 0.01$, ***$P < 0.005$, ****$P < 0.0001$

were grown in YPD (see above without the technical agar) except to prepare the inoculum for in vivo experiments where strains were grown in NGY medium (0.1% (w/v) Neopeptone (BD Biosciences), 0.4% (w/v) glucose (Fisher Scientific) and 0.1% (w/v) yeast extract (Oxoid). *Aspergillus fumigatus* clinical isolate V05–27 was cultured on Potato Dextrose Agar slants for 7 days before the spores were harvested by gentle shaking with sterile 0.1% Tween-20 in phosphate-buffered saline (PBS). Harvested spores were purified, counted and resuspended at a concentration of $1 \times 10^8$ spores/ml. Swollen spores were generated by incubation in RPMI media for 4 h at 37 °C. *Cryptococcus neoformans* KN99α and *Cryptococcus gattii* R265 were grown in YPD overnight, washed in PBS and $1 \times 10^7$ cells were added to 6 ml RPMI +10% foetal calf serum (FCS) in 6-well plates. Plates were incubated at 37 °C+5% $CO_2$ for 5 days to induce capsule formation. Harvested cells were washed in PBS. Rat lung tissue isolates of *Pneumocystis carinii* M167–6 were washed in PBS and immunostained.

**Generation of recombinant Hyr1 N-terminus fragment protein**. A recombinant N-terminal fragment of *C. albicans* Hyr1 (amino acids 63 to 350, Supplementary Figure 1a) incorporating an N-terminal 6xHis tag was expressed in HEK293F cells and purified by nickel-based affinity chromatography using a NTA superflow column (QIAGEN, USA). Fractions containing recombinant Hyr1 were pooled and further purified via Analytical Superdex 200 gel filtration chromatography (GE Healthcare, USA) in PBS. QC of the recombinant protein via SDS-PAGE gel analysis and analytical SEC confirmed a protein of a predicted mass of 32 kDa (Supplementary Figure 1b, c).

**PBMC isolation and purification of circulating IgG from donor plasma**. EDTA-containing peripheral venous blood was collected from individuals who had recovered from a *Candida* infection within the last year. Separation of peripheral blood mononuclear cells (PBMCs) and plasma from whole blood was carried out via density gradient separation using Accuspin System-Histopaque-1077 kits (Sigma-Aldrich). PBMCs were washed three times in PBS, resuspended at a concentration of $1 \times 10^7$ cells/ml in R10 media (RPMI 1640 (Gibco, Life Technologies), 10% FCS, 1 mM sodium pyruvate (Sigma), 10 mM HEPES (Gibco, Life Technologies), 4 mM L-glutamine (Sigma), 1× penicillin/streptomycin (Sigma)) containing additional 10% FCS and 10% dimethyl sulphoxide before storing in liquid nitrogen. IgG was purified from donor plasma using VivaPure MaxiPrepG Spin columns (Sartorius Stedman). Concentration of eluted IgG was measured by absorbance at 280 nm.

**Circulating antigen-specific IgG titres**. Circulating IgG was screened against target antigens via ELISA to identify donors for subsequent CSM B-cell isolation and activation. NUNC 384-well Maxisorp microtitre plates (Sigma-Aldrich) were coated with extracted *C. albicans* cell walls or *C. albicans* overnight culture (whole cell) or 1 µg/ml recombinant Hyr1 protein antigen in PBS. Following incubation at 4 °C overnight, wells were washed three times with PBS+0.05% Tween and then blocked with PBS+0.05% Tween+0.5% bovine serum albumin (BSA) for 1 h at room temperature. After three washes, titrated purified IgG or IVIG in block buffer was added in duplicate, and plates were incubated for 2 h at room temperature. Wells were washed three times before addition of goat anti-human IgG, horse-radish peroxidase conjugated (Thermo Scientific # 31413) secondary antibody at 1:5000 dilution in blocking buffer. Following incubation for 45 min at room temperature, wells were washed and stained using tetramethylbenzidine (TMB) (Thermo Scientific). Plates were incubated for 5 min before addition of 0.18 M sulphuric acid and then read at an absorbance of 450 nm. Labstats software in Microsoft Excel was used to generate concentration-response curves for $EC_{50}$ determination and donor selection for subsequent CSM B-cell isolation and activation.

**Isolation and activation of CSM B cells**. Approximately $5 \times 10^7$ PBMCs were thawed and diluted 1:10 in pre-warmed R10 medium and treated with benzonase nuclease HC, purity >99% (Novagen) at 1:10,000. PBMCs were washed twice in PBS before final resuspension in pre-warmed R10 medium. CSM B cells were isolated from PBMCs by depletion of non-target cells using a Switched Memory B cell isolation kit with Pre-Separation Filters and LS columns on a MACS Separator (MACS Miltenyi Biotec). To activate CSM B cells and promote antibody secretion, R10 medium was supplemented with a cocktail of antibodies and cytokines to make complete R10 medium. CSM B cells were resuspended in complete R10 medium at 56 cells/ml and then plated out in 90 µl aliquots (5 cells/well) in ThermoFisher Matrix 384-well plates using a Biomek FX (Beckman Coulter). Cells were incubated at 37 °C, 5% $CO_2$ for 7 days. On day 7, 30 µl/well of supernatant was replaced with 30 µl fresh complete R10. On day 13, all supernatants were harvested and screened against recombinant N-terminus Hyr1 protein antigen, *C. albicans* extracted cell walls or *C. albicans* whole cells from overnight cultures by ELISA. B-cell activation and culturing was monitored by measuring IgG1 concentrations in B-cell supernatants at day 7 and day 13.

**Screening of IgG in B-cell supernatant against target antigens via ELISA**. The same ELISA protocol was employed for the screening of B-cell supernatants against target antigens with the exception that neat B-cell supernatant was added in the place of titrated purified IgG or IVIG. Positive hits were defined as wells with an $OD_{450}$ reading >4× background. B cells in 'positive hit' wells were lysed in buffer (1 ml DEPC-treated $H_2O$ (Life Technologies), 10 µl 1 M Tris pH 8, 25 µl RNAsin Plus RNAse Inhibitor (Promega)) and stored at –80 °C.

**cDNA synthesis and PCR1**. A schematic of the cloning protocol is shown in Supplementary Figure 2. Primers used for the RT-PCR reaction were based on those used by Smith et al.[46] and are listed in Supplementary Tables 2–5. To ensure all possible VH germline families were captured during the amplification, four forward primers specific to the leader sequences encompassing the different human VH germline families (VH1–7) were used in combination with two reverse primers —both in the human IgCH1 region. For the RT-PCR of human Vκ-Cκ genes, three forward primers specific to the leader sequences for the different human Vκ germline families (Vκ1–4) were used with a reverse primer specific to the human kappa constant region (Cκ) and two further reverse primers which were specific to the C- and N-terminal ends of the 3' untranslated region (UTR). To capture the repertoire of human Vλ genes, seven forward primers capturing the leader sequences for the different human Vλ germline families (Vλ1–8) were used with two reverse primers complementary to the C- and N-terminal ends of the 3' UTR along with another reverse primer specific to the human lambda constant region (Cλ). Amplification of the variable domain of human Ig heavy chain genes (VH), the variable and constant domains of human Ig kappa light chain genes (Vκ-Cκ) and the variable and constant domains of human Ig lambda light chain genes (Vλ-Cλ) was performed in separate reactions.

Prior to complementary DNA (cDNA) synthesis, B-cell lysates were thawed and diluted 1:5, 1:15 and 1:25 in nuclease-free $H_2O$ (Life Technologies) before addition of oligo-dT$_{20}$ (50 µM) (Invitrogen, Life Technologies) and incubation at 70 °C for 5 min.

Reverse transcription and the first PCR reaction (RT-PCR) were performed sequentially in 96-well PCR plates using the QIAGEN OneStep RT-PCR kit, with gene-specific forward and reverse primer mixes (10 µM) and neat or diluted (1:5, 1:15, 1:25) B-cell lysate as the template.

**Amplification of VH, Vκ-Cκ and Vλ-Cλ genes: nested PCR reaction**. Primers used for the nested PCR reaction were again based on those used by Smith et al.[46]. A total of 27 forward primers specific for the VH framework 1 (FW1) sequence were used together with two reverse primers specific for the framework 4 (FW4)

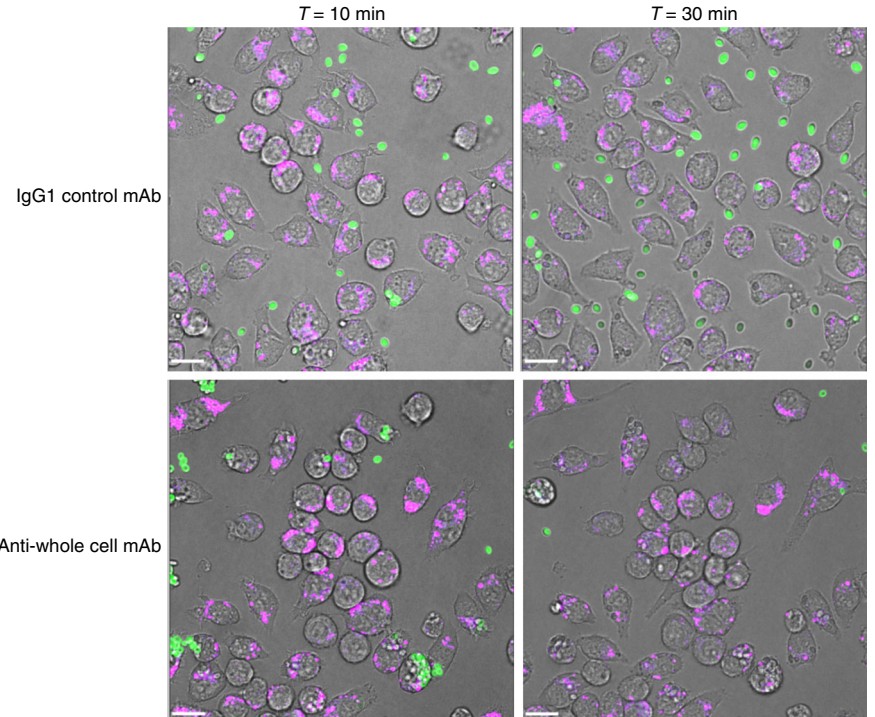

**Fig. 9** Effect of anti-*Candida* mAbs on macrophage phagocytosis of live *C. auris* cells. Snapshot images taken from live cell video microscopy capturing *C. auris* phagocytosis by J774.1 macrophages at $T = 10$ min and $T = 30$ min following pre-incubation of *C. auris* cells with IgG1 control antibody or anti-whole cell reactive antibody. Macrophages (magenta), *C. auris* (green). The scale bars represent 18 μm

region of the VH gene. For nested PCR of the Vκ-Cκ gene, a mixture of 18 forward primers specific for human Vκ FW1 sequence were used with a reverse primer specific to the human kappa constant region 3' end. For amplification of the Vλ-Cλ gene, a mixture of 31 forward primers specific for human Vλ FW1 sequences were used together with a reverse primer that was placed at the 3' end of the human lambda constant region. All of these primers contained 15 bp extensions which were complementary to the target downstream pTT5 expression vector. Nested PCR reactions were carried out using Platinum PCR SuperMix High Fidelity (Invitrogen, Life Technologies), nested gene-specific forward (10 μM) and reverse (10 μM) primer mixes and cDNA template from the RT-PCR reaction. PCR amplifications of VH genes, Vκ-Cκ genes and Vλ-Cλ genes were carried out in separate reactions and then stored on ice. After the nested PCR reaction, samples were analysed via agarose gel electrophoresis and positive hits identified and taken forward for downstream In-Fusion cloning with pTT5 mammalian expression vector.

**pTT5 mammalian expression vector preparation**. pTT5 was the expression vector used for recombinant mAb expression (licensed from the National Research Council of Canada (NRCC))[63]. pTT5 vector plasmid containing an IgG1 heavy chain gene in the multiple cloning site was linearised by double digestion using FastDigest Restriction enzymes (Thermo Scientific) in separate reactions to facilitate generation of HC and LC backbones for subsequent cloning of VH and Vκ-Cκ or Vλ-Cλ genes. HC and LC backbone DNA was run on a 1% agarose gel and bands were excised from the gel and purified using the QIAquick Gel Extraction kit (QIAGEN). DNA was quantified on a NanoVue Plus Spectrophotometer (GE Healthcare). The 3'- and 5'-termini of the linearised plasmids were dephosphorylated using FastAP Thermosensitive Alkaline phosphatase (Thermo Scientific) to prevent vector self-ligation. Reaction mixtures were cleaned using the MinElute Reaction Cleanup Kit (QIAGEN) and then run on 1% agarose gels. Bands corresponding to dephosphorylated HC and LC backbones were excised from the gel and purified using the QIAQuick Gel Extraction kit (QIAGEN) as above. Dephosphorylated linearised vector DNA was quantified on a NanoVue Plus spectrophotometer (GE Healthcare).

**In-Fusion cloning**. The In-Fusion HD Cloning Kit (Clontech, USA) was used to clone the VH, Vκ-Cκ and Vλ-Cλ genes into the pTT5 mammalian expression vector before transformation of Stellar Competent cells in a 96-well plate format (Clontech). Transformed cells were recovered in SOC medium (Clontech) with shaking at 37 °C for 45–60 min before plating out onto LB agar plates (1% (w/v) tryptone, 0.5% (w/v) yeast extract, 1% (w/v) NaCl, 1.5% (w/v) agar) containing 100 μg/ml ampicillin. Single colonies picked for inoculation of 2xTY media containing 100 μg/ml ampicillin in a Greiner deep well, 96-well plate (Sigma). Plasmid

miniprep DNA was isolated from cultures in 96-well microtitre plates and an epMotion® 5075 laboratory robot (Eppendorf, Germany) and stored at –20 °C until required for mammalian transfections following sequence analysis of complementary determining region (CDR) diversity and comparison to germline sequences.

**Small- and large-scale expression of recombinant mAbs**. A file containing all possible VH and Vκ/Vλ combinations resulting from the original hit wells from the primary ELISA screen was generated. Automated mixing of native HC and LC DNA pairing combinations (1.5 μg of HC plasmid DNA and 1.5 μg of LC plasmid DNA) was facilitated using a Hamilton Microlab® Starline liquid handling platform (Life Science robotics, Hamilton Robotics). Transient transfections of 3 ml of cultured Expi293F HEK cells (Gibco, USA) was at a density of $2.5 \times 10^6$ cells/ml in 24-well tissue culture plates using the Expifectamine 293 Transfection kit (Life Technologies, USA). Expi293F cells were maintained in sterile Expi293 expression media (Invitrogen) without antibiotics at 37 °C, with 7% CO$_2$, 120 rpm shaking. For downstream large-scale transfections, DNA was prepared using a QIAGEN Plasmid Maxi Kit (QIAGEN, USA) with typical yields of 1.5 μg/μl. Large-scale transfections were carried out using 100 μg of total DNA (50 μg of HC plasmid DNA and 50 μg LC plasmid DNA) and a 100 ml of suspension cultured Expi293F cells (Life Technologies). Supernatants were harvested on day 6 and recombinant mAb expression was quantified using anti-human IgG Fc sensors on an Octet QK$^e$ (ForteBio, CA, USA). Following upscaling, recombinant mAb expression was quantified with an Octet before purification via affinity-based FPLC using HiTrap Protein A HP columns on an ÄKTA (GE Healthcare). mAbs were eluted in 20 mM citric acid, 150 nM NaCl (pH 2.5) before neutralisation with 1 M Tris buffer (pH 8) and then dialysis in PBS overnight. IgG concentration was quantified on a NanoVue Spectrophotometer (GE Healthcare).

**QC of recombinant mAbs**. Purified recombinant mAbs were checked via SDS-PAGE gel analysis using 4–12% Bis-Tris SDS-PAGE gels under reducing and non-reducing conditions, analytical SEC and analytical mass spectrometry. Confirmation of binding to original target antigen was carried out via ELISA using the protocol described for the circulating antigen-specific IgG screen.

**Immunofluorescence imaging of mAbs binding to fungal cells**. Single colonies of *Candida* were inoculated into 10 ml YPD medium and incubated at 30 °C, 200 rpm overnight. Cultures were diluted 1:1333 in milliQ water and then adhered on a poly-L-lysine-coated glass slide (Thermo Scientific, Menzel-Gläser) for 30 min. To induce filamentation, cells were incubated in pre-warmed RPMI+10% FCS at 37 °C for 90 min to 2 h (this step omitted for staining of yeast cells). For *A. fumigatus*, hyphal growth was induced from swollen conidia grown in RPMI+10% FCS. Other

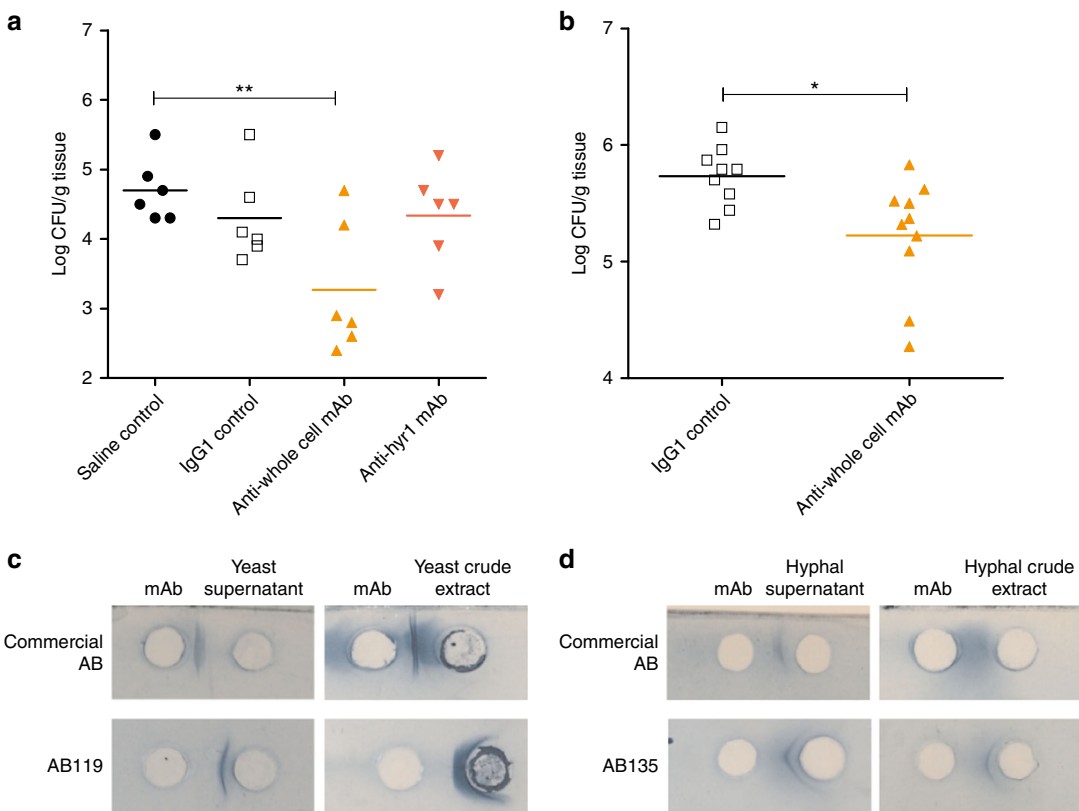

**Fig. 10** Assessment of anti-*Candida* mAbs in murine models of disseminated candidiasis and a routine diagnostic assay. **a** *C. albicans* SC5314 was pre-incubated with saline, IgG1 control, anti-whole cell mAb (AB119) or anti-Hyr1 mAb (AB120) and then injected i.v. into the tail vein of female BALB/c mice ($n = 6$ per group). Kidney fungal burdens from each group were determined on day 3 post infection. **b** IgG1 control or anti-whole cell mAb (AB119) was administered i.p. 4 h prior to injection of *C. albicans* SC5314 i.v. into the lateral tail vein of male CD1 mice ($n = 10$ per group). Kidney fungal burdens from each group were determined on day 7 post infection. Dots represent individual animals and horizontal lines represent mean; statistical significance was determined by two-tailed *t*-test; *$P < 0.05$, **$P < 0.01$. Purified anti-*Candida* mAbs react with yeast (**c**) and hyphal (**d**) antigenic preparations in a counterimmunoelectrophoresis assay routinely employed to diagnose *Candida* infection. Precipitin lines between wells indicate a positive reaction

fungal cells were prepared as described in fungal strains and growth conditions. All fungal cells were then washed in Dulbecco's phosphate-buffered saline (DPBS) and fixed with 4% paraformaldehyde, washed and blocked with 1.5% normal goat serum (Life Technologies) before staining with an anti-*Candida* mAb at 1–10 μg/ml for 1 h at room temperature. After three washes with PBS, cells were stained with Alexa Fluor® 488 goat anti-human IgG antibody (Life Technologies #A11013) at a 1:400 dilution and incubated at room temperature for 1 h prior to imaging in three dimensions (3D) on an UltraVIEW® VoX spinning disk confocal microscope (Nikon, Surrey, UK).

**High-pressure freezing of *C. albicans* cell samples**. *C. albicans* yeast and hyphal cell samples were prepared by high-pressure freezing using an EMPACT2 high-pressure freezer and rapid transport system (Leica Microsystems Ltd., Milton Keynes, UK). Using a Leica EMAFS2, cells were freeze-substituted in acetone+1% (w/v) OsO4 before embedding in Spurr's resin and polymerising at 60 °C for 48 h. A Diatome diamond knife on a Leica UC6 ultramicrotome was used to cut ultrathin sections which were then mounted onto nickel grids.

**Immunogold labelling of samples for transmission electron microscopy**. Sections on nickel grids were blocked in blocking buffer (PBS+1% (w/v) BSA and 0.5% (v/v) Tween20) for 20 min before incubation in incubation buffer (PBS+0.1% (w/v) BSA) for 5 min times 3. Sections were then incubated with anti-*Candida* mAb (5 μg/ml) for 90 min before incubation in incubation buffer for 5 min, for a total of 6 times. mAb binding was detected with Protein A conjugated to 10 nm gold (Aurion) (diluted 1:40 in incubation buffer) for 60 min before another six, 5 min washes, in incubation buffer, followed by three, 5 min washes in PBS and three, 5 min washes in water. Sections were then stained with uranyl acetate for 1 min before three, 2 min washes, in water and left to dry. TEM images were taken using a JEM-1400 Plus using an AMT UltraVUE camera.

**Enzymatic modification of *C. albicans* cell wall**. For proteinase K treatment, single colonies of *Candida* were inoculated into 10 ml YPD medium and incubated at 30 °C, 200 rpm overnight. Cultures were diluted in milliQ water and then adhered on poly-L-lysine-coated glass slides. To induce filamentation, cells were incubated in pre-warmed RPMI+10% FCS at 37 °C for 90 min to 2 h. Slides were washed with DPBS and cells were treated with 50 μg/ml proteinase K at 37 °C for 1 h. For Endo-H, α(1–2,3,6)-mannosidase from Jack Bean and Zymolyase 20 T treatments, *C. albicans* overnight yeast cells were washed and resuspended in DPBS. Filamentous cells were induced as above. Cells were washed in DPBS and resuspended in Glycobuffer and Endoglycosidase H (10 U/μl; NEB), 20 mM sodium acetate with 0.4 mM $Zn^{2+}$ pH 5 and α-(1–2,3,6)-mannosidase or 20 mM PIPES with 2 M Sorbitol pH 6.5 and Zymolyase 20T (50 U/g wet cells; MPBIO) at 37 °C for 2 h. Control cells were kept in reaction buffers without enzymes. Cells were then washed in DPBS and fixed with 4% paraformaldehyde, washed and blocked with 1.5% normal goat serum (Life Technologies) before staining with an anti-*Candida* mAb at 1 μg/ml for 1 h at room temperature. After 3 washes with DPBS, cells were stained with Alexa Fluor® 488 goat anti-human IgG antibody (Life Technologies) at a 1:400 dilution and incubated at room temperature for 1 h prior to imaging in 3D on an UltraVIEW® VoX spinning disk confocal microscope (Nikon, Surrey, UK).

**Preparation of human monocyte-derived macrophages**. Human macrophages were derived from monocytes isolated from the blood of healthy volunteers. PBMCs were resuspended in Dulbecco's modified Eagle's medium (DMEM) (Lonza, Slough, UK) supplemented with 200 U/ml penicillin/streptomycin antibiotics (Invitrogen, Paisley, UK) and 2 mM L-glutamine (Invitrogen, Paisley, UK). Serum isolated from blood was heat inactivated for 20 min at 56 °C. PBMCs were seeded at $6 × 10^5$ in 300 μl/well supplemented DMEM containing 10% autologous human serum, onto an 8-well glass-based imaging dish (Ibidi, Munich, Germany) and incubated at 37 °C with 5% $CO_2$ for 1 h 45 min to facilitate monocyte adherence to the glass surface. Floating lymphocytes in the supernatant were aspirated and the same volume of fresh pre-warmed supplemented DMEM containing 10% autologous human serum added to the well. Cells were incubated at 37 °C, 5% $CO_2$ for 7 days with media changed on days 3 and 6[64]. Cells were used in imaging experiments on day 7. Supplemented DMEM was replaced with pre-warmed supplemented $CO_2$-independent media containing 1 μM LysoTracker Red DND-99 (Invitrogen) immediately prior to phagocytosis experiments.

**Glycan microarray screening analysis**. For microarray screening analysis, two microarray platforms were used: (1) an array, designated 'Fungal, Bacterial and Plant Polysaccharide Array' featuring 19 saccharides (polysaccharides or glyco-proteins) derived from fungi, bacteria and plants and one lipid-linked neoglyco-lipid (NGL) probe derived from the hexasaccharide of β1,4-linked N-acetylglucosamine (GlcNAc) (representative of fungal chitin), and (2) an array of 38 sequence-defined NGL probes of N-glycans (majority of mammalian-type) (Supplementary Table 8) designated 'N-glycan Array Set 3', described previously[65]. The polysaccharides and glycoprotein antigens included in the Fungal, Bacterial and Plant Polysaccharide Array are listed in Supplementary Table 7. The glucan polysaccharides (IDs 1–11) have been described previously[66]. The mannan from S. cerevisiae (ID 12) was purchased from Sigma. Purified Candida albicans N-linked mannoprotein preparation (ID 13)[67] was a kind gift from David Williams (East Tennessee State University); the purified Aspergillus fumigatus mannoprotein preparation (ID 14)[25] was a kind gift from Christopher Thornton (University of Exeter). The antigen preparations from Mycobacterium smegmatis and M. tuber-culosis (IDs 15–18) were obtained from the National Institutes of Health (NIH) Biodefense and Emerging Infections Research Resources Repository (BEI Resour-ces) and were described previously[68]. The glucurono-xylomannan from Tremella fuciformis (ID 19) was purchased from Elicityl. For construction of the microarrays the saccharides and the NGL probes were immobilised noncovalently on nitrocellulose-coated glass slides, following established protocols[53,66]. Poly-saccharides and glycoproteins were taken up in water, with the exception of cur-dlan polysaccharide that was solubilised using mild alkaline solution (50 mM NaOH) and glucurono-xylomannan solubilised in 150 mM NaCl. The microarrays were probed with the anti-Candida mAbs, following described protocols[53,66]. In brief, after blocking with 10 mM HEPES-buffered saline (pH 7.4), 150 mM NaCl, 5 mM CaCl₂ (referred to as HBS) containing 1% w/v BSA (Sigma) and 0.02% v/v Casein (Pierce), the microarrays were overlaid with the mAbs diluted to a final concentration of 10 μg/ml (for the Fungal, Bacterial and Plant Polysaccharide Array) and at 50 μg/ml (for the N-glycan Array Set 3) in the blocking solution. Binding was detected with biotinylated anti-human IgG (Vector, 5 μg/ml) fol-lowed by Alexa Fluor-647-labelled streptavidin (Molecular Probes, 1 μg/ml), dilu-ted in the blocking solution. The mAb PGT 128 was a kind gift from Katie J. Doores (King's College London, UK) and was analysed at 50 μg/ml[55]. Fc-dectin-1 was kindly provided by Gordon Brown (University of Aberdeen, UK) and was analysed at 20 μg/ml in the blocking solution 0.5% v/v casein (Pierce) in HBS[66]. Analyses were performed at ambient temperature. Microarray data analysis was performed using dedicated software developed by Mark Stoll of the Glycosciences Laboratory[69].

**FACS**. Candida and S. cerevisiae colonies were grown in YPD medium and incubated at 30 °C, 200 rpm overnight. Cells were washed twice in 1× PBS, counted and then centrifuged and fixed in 4% paraformaldehyde for 45 min. Cells were then washed twice in 1× PBS to remove residual paraformaldehyde before $2.5 \times 10^6$ cells/well were added into V-bottomed 96-well tissue culture plates. Anti-Candida mAb was then added to wells at 1 μg/ml in FACS buffer (1× PBS, 1% FBS, 0.5 mM EDTA) and incubated for 45 min. Cells were washed once with FACS buffer and then stained with 5 μg/ml Alexa Fluor® 488 goat anti-human IgG antibody (Life Technologies) for 30 min at room temperature. Stained cells were then washed twice in FACS buffer before final resuspension in FACS buffer and storage in the dark at 4 °C. Samples were analysed on a BD Fortessa flow cytometer where 10,000 events were acquired for each sample from 3 independent experiments. Median fluorescence intensity was calculated for each sample using FloJo v.10 software. The gating strategy employed is shown in Supplementary Figure 6.

**Preparation of J774.1 mouse macrophages**. J774.1 macrophages (ECACC, HPA, Salisbury, UK) were maintained in tissue culture flasks in DMEM (Lonza) sup-plemented with 10% (v/v) FCS (Biosera, Ringmer, UK), 200 U/ml penicillin/ streptomycin antibiotics (Invitrogen) and 2 mM L-glutamine (Invitrogen) and incubated at 37 °C, 5% CO₂. For phagocytosis assays, macrophages were seeded in supplemented DMEM at a density of $1 \times 10^5$ cells/well in an 8-well glass-based imaging dish (Ibidi) and incubated overnight at 37 °C, 5% CO₂. Immediately prior to phagocytosis experiments, supplemented DMEM was replaced with pre-warmed supplemented CO₂-independent media (Gibco) containing 1 μM LysoTracker Red DND-99 (Invitrogen).

**Preparation of FITC-stained C. albicans**. C. albicans colonies were grown in YPD medium and incubated at 30 °C, 200 rpm overnight. Live C. albicans cells were stained for 10 min at room temperature in the dark with 10 mg/ml FITC (Sigma, Dorset, UK) in 0.05 M carbonate-bicarbonate buffer (pH 9.6) (BDH Chemicals, VWR International, Leicestershire, UK). Following 10 min of incubation, in pha-gocytosis assays using C. albicans FITC-labelled yeast, the cells were washed three times in 1× PBS to remove any residual FITC and finally resuspended in 1× PBS or 1× PBS containing purified anti-Candida mAb at 1–50 μg/ml. For assays where filamentous C. albicans cells were added to immune cells, these were washed and resuspended in supplemented CO₂-independent media with or without anti-Candida mAb at 1–50 μg/ml and incubated at 37 °C with gentle shaking for 45 min.

**Live cell video microscopy phagocytosis assays**. Phagocytosis assays were performed using a standard protocol with modifications[64,70,71]. Following pre-incubation with/without 50 μg/ml anti-Candida mAb, live FITC-stained wild-type C. albicans (CA14-CIp10) yeast or hyphal cells were added to LysoTracker Red DND-99-stained J774.1 murine macrophages in 8-well glass-based imaging dish (Ibidi) at a multiplicity of infection of 3. For experiments investigating the effect of a mFcγR blocker on the phagocytosis of C. albicans pre-incubated with 5 μg/ml anti-Candida whole cell mAb (AB140), J774.1 macrophages were incubated with 10 μg/ml purified rat anti-mouse CD16/CD32 mAb (BD Pharmingen, BD Bios-ciences #553142) for 5 min at room temperature before addition of C. albicans cells. Video microscopy was performed using an UltraVIEW® VoX spinning disk confocal microscope (Nikon) in a 37 °C chamber and images were captured at 1 min intervals over a 3 h period. At least three independent experiments were performed for each antibody. Twenty-five macrophages were selected at random from each experiment and analysed individually at 1 min intervals over a 3 h period using Volocity 6.3 imaging analysis software (Improvision, PerkinElmer, Coventry, UK). Measurements included C. albicans uptake—defined as the number of C. albicans cells taken up by an individual phagocyte over the 3 h period—and C. albicans rate of engulfment—defined as the time point at which cell–cell contact was established until the time point at which C. albicans was fully engulfed. Finally, Volocity 6.3 imaging analysis software was used to measure the distance travelled, directionality and velocity of macrophages at 1 min intervals until the first uptake event to provide a detailed overview of macrophage migration towards C. albicans cells. Mean values and standard errors of the mean were calculated. One-way analysis of variance (ANOVA) followed by Bonferroni multiple comparison tests, Kruskal–Wallis test with Dunn's multiple comparisons test or unpaired, two-tailed t-tests with Welch's correction were used to determine statistical significance.

**Proof of concept in 3-day systemic candidiasis infection model**. A 3-day model of disseminated candidiasis was employed to assess the efficacy of anti-Candida mAbs in vivo in immunocompetent 6–8-week-old female BALB/c mice (Envigo, UK)[56]. To determine the minimum group size required to detect a statistical dif-ference in outcomes ($P < 0.05$), power analyses were carried out using data gen-erated from other experiments using the same model (power = 0.8). Mice were randomly assigned to groups ($n = 6$) and were housed in individually ventilated cages (IVC), with food and water provided ad libitum. Each cage was randomly assigned a treatment, and investigators were not blinded when assessing outcome. Endotoxin levels in the purified recombinant mAb samples were determined with the Limulus Amebocyte Lysate QCL-1000 test (Lonza) to ensure that endotoxin was <0.6 EU/ml before in vivo administration. On day 0, $\sim 3.2 \times 10^5$ C. albicans SC5314 yeast cells were pre-incubated with either sterile saline, 1.5 mg/ml purified IgG1 isotype control mAb, 1.5 mg/ml purified anti-Candida mAb AB119 or 1.5 mg/ml purified anti-Hyr1 mAb AB120 in 700 μl. Following 60 min of incubation at room temperature, mice were intravenously (i.v.) infected with 100 μl via the lateral tail vein. This is the equivalent of approximately $1.6 \times 10^4$ colony-forming units (CFUs)/g per mouse, treated with 7.5 mg/kg purified antibody. Mice were mon-itored and weighed at least once per day from day 0 up to and including day 3. On day 3, animals were culled by cervical dislocation and the kidneys harvested for quantitative analysis of fungal burden.

**Proof of concept in prophylactic treatment model of systemic candidiasis**. This prophylactic treatment model was conducted by Evotec Ltd (UK). In this model, 7–8-week-old immunocompetent male CD1 mice (Charles River) were randomly assigned to groups ($n = 10$) and housed in IVC, with food and water provided ad libitum. Mice were injected i.v. via the lateral tail vein with $2.13 \times 10^5$ C. albicans SC5314 per mouse in 200 μl. At 4 h prior to challenge, each mouse received 1 mg of IgG1 control or AB119 antibody by intraperitoneal (i.p.) injection in 200 μl. Mice were monitored and weighed at least once per day from day 0 up to and including day 7. On day 7, animals were killed by administration of pento-barbitone overdose and the kidneys harvested for quantitative analysis of fungal burden.

**Analysis of disease progression**. Kidneys were homogenised and serial dilutions plated on Sabouraud dextrose agar plates (1% mycological peptone (w/v), 4% glucose (w/v), 2% agar (w/v)) and incubated at 35 °C overnight. Fungal burdens were expressed as log CFUs/g of infected organ. Weight change was measured and expressed as a percentage of mouse starting weight. Statistical analyses of fungal burdens were carried out using unpaired two-tailed t-test. An F-test was used to show there was no statistical significance in variance within the groups ($P = 0.1353$ and $P = 0.0817$ for the 3-day and 7-day models, respectively). All statistical ana-lyses were carried out using GraphPad Prism 5.04 software.

**Counterimmunoelectrophoresis**. Agar gels were prepared (Veronal buffer+0.5% (w/v) purified agar+0.5% (w/v) LSA agarose+0.05% (w/v) sodium azide, pH 8.2) and wells were cut out using a cutter. Into one column of wells, 10 μl of neat anti-Candida mAb was added. The same volume of antigen (crude C. albicans yeast or hyphal preparation (following glass bead disruption of cells and 1 min cen-trifugation at $15{,}200 \times g$ to generate disrupted cell wall/glass bead slurry and cell supernatant antigenic preparations)) was added to the second column of wells and

gels were placed into an electrophoresis tank containing veronal buffer. Gels were oriented so that the antibody wells were lined up alongside the anode and the antigen wells alongside the cathode due to antibody migration towards the cathode via electroendosmosis and antigen migration towards the anode due to lower isoelectric points than the buffer pH. The gels were run at 100 V for 90 min before removal and immersion in saline-trisodium citrate overnight. The following day, the gels were rinsed with water and covered with moistened filter paper and left to dry in an oven for 2 h. Once dried, the filter paper was moistened and removed and the gels put back into the oven for a further 15 min to dry completely. Gels were then immersed in Buffalo Black solution (0.05% (v/v) Buffalo Black, 50% (v/v) distilled water, 40% (v/v) methylated spirit, 10% (v/v) acetic acid) for 10 min before destaining in 45% (v/v) industrial methylated spirits, 10% (v/v) acetic acid, 45% (v/v) distilled water for 10 min. Gels were then dried and examined for the formation of precipitin lines.

**Statistical analysis**. Statistical analyses of data were carried out using GraphPad Prism 5.04. Results are expressed as mean ± SEM. When comparing two groups, unpaired two-tailed $t$-tests were performed with Welch's correction when appropriate. For comparison of more than two groups, one-way ANOVA with Bonferroni's post hoc comparisons or the Kruskal–Wallis test with Dunn's multiple comparisons test was performed.

## Data availability

All relevant data are available from the authors upon request.

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

## Acknowledgements

We thank the BBSRC, SULSA BioSKAPE and Pfizer Inc. for funding for a studentship for F.M.R. and the Wellcome Trust (086827, 075470, 099215, 099197 and 101873) and a Wellcome Trust ISSF award (105625), MRC CiC (MC_PC_14114) and MRC Centre for Medical Mycology and University of Aberdeen for funding and a Wellcome Trust Strategic Award (097377) and a Wellcome Trust grant 099197MA to T.F. and FCT Investigator IF/00033/2012 and PTDC/QUI-QUI/112537/2009 to A.S.P. We thank Ian Broadbent, Angus McDonald and Ron Gladue for constructive discussions; Chris Boston and Amanda Fitzgerald for advice on antibody expression and purification; Ed Lavallie and Wayne Stochaj for design and expression of the recombinant Hyr1; Louise Walker for high-pressure freezing of samples for TEM analysis; Jeanette Wagener for endotoxin testing of mAbs for in vivo experiments; Yan Liu of the Glycosciences laboratory for insight in the analysis with N-glycan array; Rebecca Hall and Mark Gresnigt for providing fungal strains; Andrew Limper and Theodore J. Kottom for providing Pneumo-cystis infected lung tissue extracts; David Williams for *C. albicans* mannoprotein; Christopher Thornton for *A. fumigatus* mannoprotein; Katie J. Doores for mAb PGT 128; and Gordon Brown for the murine Fc-Dectin-1. We are grateful to Lucinda Wight, Debbie Wilkinson and Kevin MacKenzie in the Microscopy and Histology Core Facility (Aberdeen University) and Raif Yuecel in the Iain Fraser Cytometry Centre (Aberdeen University) for their expert help with microscopy and cytometry experiments. We are also grateful to the staff at the University of Aberdeen Medical Research Facility for assistance with in vivo experiments and members of the Glycosciences Laboratory for their support of the Carbohydrate Microarray Facility.

## Author contributions

F.M.R., N.A.R.G., L.P.E. and A.J. designed research. F.M.R., N.A.R.G., L.P.E., A.J., H.W., S.E. and E.M.J. conceived and designed experiments. F.M.R., I.R., R.B., D.M.M., L.M.S., A.S.P. and T.F. carried out experiments and analysed data. F.M.R., N.A.R.G., L.P.E., A.J., L.M.S., A.S.P. and T.F. contributed to writing of the manuscript. N.A.R.G., L.P.E. and A.J. supervised the project. All authors reviewed the manuscript.

## Additional information

**Competing interests:** F.M.R., N.A.R.G., L.P.E. and A.J. are listed as inventors in a related patent application (Antibody molecules and uses thereof, PCT/GB2016/050577). The remaining authors declare no competing interests.

