## [Peer Review File · Nature Communications]

Reviewers' comments:

Reviewer #1 (Remarks to the Author):

This is a well written and well documented presentation of the generation and characterization of human B cell-derived monoclonal antibodies directed against *Candida*, and I have focused my comments on the glycan microarray analyses included in this manuscript as requested by the editor.

The glycan microarray analysis was carried out by an extremely competent group with many years of experience in developing and analyzing glycan microarrays. The descriptions and documentation of the methods for printing the and analyzing the two glycan arrays mentioned are adequate and the references included cover the details of generating the targets that were printed and analyzed. That being said, I would like to make sure authors and editors are aware of an initiative, designated MIRAGE (Minimum Information Required for A Glycomics Experiment), which was created by experts in the fields of glycobiology, glycoanalytics and glycoinformatics to produce guidelines for reporting results from the diverse types of experiments and analyses used in structural and functional studies of glycans in the scientific literature. It would be appropriate and informative for investigators reading the manuscript if the authors included a statement in this manuscript simply stating that the glycan microarray studies followed the guidelines as published by the MIRAGE initiative (Glycobiology, 2017, vol. 27, no. 4, 280–284).

The microarray data were used to demonstrate the specificity of the antibodies for *C. albicans* mannoprotein and the absence of binding to other fungal or bacterial glycans presented on the array. The “validation” of the arrays were done using Dectin-1 for the Fungal, and Bacterial Polysaccharide Array (demonstrating the presence of beta1,3-glucans) and antibody PGT128 for the N-Glycan Array Set 3 (demonstrating the presence of the Man8 and Man9 N-glycans). This group would certainly have used other well-defined glycan binding proteins to demonstrate that the “probes” were printed and available for binding; these additional validation indicators should be mentioned in the supplemental information. It would be useful to mention something regarding the statistical analyses of the results and the number of replicates of each “probe” on the arrays. This is not as critical since there were only a few positive results with large values relative to background.

It was not clear how representative mAbs were defined (page 9, line 196). There were 17 mAbs developed and 5 of them appeared directed against the Hyr1 recombinant protein and the remaining 12 bound *C. albicans* whole cells. AB121 was the only Hyr1-binding antibody tested on the two arrays, and only 7 of the other 12 that were candidate for anti-glycan binding activity were tested. The rationale for the selection could be included.

Finally, it interesting that AB135 binding to *C. albicans* was reduced by zymolase suggesting this antibody might be specific for a beta1,3-glucan, but beta1,3-glucans on the array were not bound by this antibody (Fig. 4a). The authors should comment on this.

Minor points:

Page 24 line 606–612 - The methods section described the two designated glycan arrays as “Fungal, and Bacterial Polysaccharide Array” and the “N-Glycan Array Set 3”. These arrays are defined in Supplementary Tables 3 and 4, but the designated titles are not actually stated in the Table legends (Table 3 legend did have a version of the designation mentioned).

Page 9 line 201 – The *C. albicans* mannoprotein has an ID number of ID-13, but the text should indicate which array is being referred to.

Reviewer #2 (Remarks to the Author):

This paper reports on the species-specific and pan-Candida human recombinant mAbs displayed properties for diagnostics and generated strong opsono-phagocytic activity of macrophages in vitro, and were protective in a murine model of disseminated candidiasis. Fully human antibodies would represent highly valuable reagents to explore future immunotherapies targeting medical mycoses, therefore, the manuscript is important to scientists in the specific field, and report on "human antibodies that target the major human pathogen *Candida* spp and have therapeutic and diagnostic potential" are novel. While I think this paper is of interest to those active in this important field, I have some comments requiring author's clarification and some concern about both the results and the conclusions.

1. The author mentioned that human antibody encoding V genes targeting *Candida* epitopes were cloned from single B cells derived from donors who had recovered from mucosal *Candida* infections. Actually, to select potential B cell clones producing antibodies related to protection / good prognosis, is it better to select donors who had recovered from systemic candidiasis? Please explain why the mucosal infection patients were selected.
2. Only *Candida albicans* and *C. dubliniensis* are truly polymorphic, due to their ability to form hyphae and/or pseudohyphae. Except *Candida albicans*, is the author aware of any other non-*albicans* *Candida* species have *Hyr1* gene?
3. The author's *Hyr1* related mAbs are valuable for *C. albicans* identification, however, antibodies with pan-species activity may not have much translational potential as diagnostics, because laboratory diagnosis has improved with the advent of new methods for *Candida* isolation and species identification. Rapid identification of *Candida* species has become more important because of an increase in infections caused by species other than *Candida albicans*, including species innately resistant to traditional antifungal drugs. For example, *Candida auris*, an emerging fungus that can cause invasive infections, is associated with high mortality and is often resistant to multiple antifungal drugs. Early species identification is critical for the clinical effectiveness of antimicrobial treatment.
4. The author has shown nice data of "macrophage phagocytosis of live *C. albicans* cells pre-incubated with mAbs"; it is also important to determine macrophage candidacidal activity, especially in the presence of *hyr1*-specific mAbs. Filaments of *C. albicans* are required for tissue damage and escape killing by macrophages. Filamentous forms (hyphae and/or pseudohyphae) of *Candida* species also demonstrate increased resistance to phagocytosis compared with yeast. It will be interesting to show colony forming units (CFU) after 24[^]36 hour incubation at 37C with macrophages pre-incubated with *hyr1*-specific mAbs.
5. The author concluded that macrophages travelled faster and further towards *C. albicans* yeast cells when pre-incubated with an anti-whole cell mAb, did the author observe the same - macrophages travelled faster and further -when pre-incubated with *Hyr1*- mAbs? If yes, what is the mechanism?
6. Please further explain why the author use a three-day mouse model of disseminated candidiasis to assess the protective efficacy of mAb in vivo. I understand it's a new novel of invasive *C. albicans* infection of mice, and changes associated with disease become measurable within 3 days of challenge with *C. albicans*. However, evaluation of virulence effects solely in terms of kidney burdens and outcome scores seems a rather crude and unsophisticated approach. It will be interesting to see the differences in survival/ mean survival time. Furthermore, regard to two targeted organs in mouse candidiasis model mimic humans, fungal burden in the brain usually peaks on day 4 and then declined by day 7, whereas the kidney fungal burden continues to

increase inexorably, reach peak by day 7.

7. Please clarify what is challenge dose for each mouse in systemic disseminated candidiasis model. The Author mentioned 1×10^4 CFU/g per mouse, does it mean 1×10^4 *C. albicans* cells for each mouse or the number have to be multiplied by weight of each mouse? It's confusing. For BALB/c mice, generally when challenged with lethal dose of *C. albicans* cells (5×10^5 per mouse), in fact the controls (all moribund) have about 10^7 to the seventh kidney counts and all die within 5-7 days. At fungal burden of log 4-5 of control group in this manuscript, usually mice survive well for a period of time.

In addition, the differences in kidney fungal burden among control and mAb-treated group are not impressive, only 1-1.5 log differences. The protective efficacy of anti-whole cell mAbs will be more convincing if evidenced by both kidney CFU and survival during extended time (2-3 weeks). In mouse model of disseminated candidiasis, many research groups demonstrated that vaccinations could result in marked improvement in survival and significant reductions in fungal burden during otherwise rapidly fatal hematogenously disseminated candidiasis in both immunocompetent and immunocompromised mice. Of interest are the kidney fungal burden from vaccinated mice could be under 2-3 log CFU/g. Generally, mice with kidney fungal burden indicative of a fatal infection is 6-7 log CFU/g; mice with kidney fungal burdens above this level typically die from infection, whereas mice with kidney fungal burdens below 4-5 log CFU/g could survive the infection. The conclusions of the manuscript that the species-specific and pan-*Candida* mAbs were protective in a murine model of disseminated candidiasis is not convincing.

Reviewer #3 (Remarks to the Author):

This is a very interesting manuscript which takes antibody technology to a new level in the potential management of invasive mycoses. In a set of robust and creative experiments this work uses the isolation of single class switched memory B cells isolated from donors serum-positive for anti-*Candida* IgG and screened them for recognition of *hyr1* cell wall protein and a whole cell wall preparations. The reactive antibody genes(s) were cloned and expressed in kidney cells to make specific recombinant anti-*Candida* monoclonal antibody. A pan *Candida* mAbs then was shown to have opsonic activity and appeared to have protective features in a murine model of disseminated candidiasis.

The very strong features of this strategy are:

(1) the creative technology to identify and create the potential protective mAbs. Clearly, this work showed a nice technology to keep the discovery of humanized antibodies for antifungal therapy alive and well which will be essential for future therapeutics.

(2) By targeting *Candidiasis*, the investigators have not only focused on a major fungal pathogen which could use both new treatment strategies but also preventive strategies. Furthermore, there is a rich and unfulfilled history of antibody treatment in *Candidiasis* with the development of Mycograb® which had a positive therapeutic signal in human disease until its development was stopped.

(3) The manuscript is clearly written and the story is easy to follow in this presentation. There are a series of methodological maneuvers but they are explained well and need to be discussed to appreciate the value of this technology. I have little to remark on the strategy and creation of the monoclonal antibodies. Well done!

The primary issues that I would like investigators to address are the following:

(1) It would be helpful to understand why B-cells were taken from antibody positive mucosal-infected patients. It just seemed like critically formed antibodies for treatment and prevention of invasive disease would come from patients who recovered from a candidemia and/or internal

invasive candida disease. Are B-cells producing antibodies during mucosal disease the same as those responding to invasive disease? Are they as potent?

(2) The elegance of the B-cell and antibody technology (cloning and screening) seems to be somewhat dampened by more meager animal study endpoints for efficacy. There are two issues the investigators should address.

(1) It appears that the use of these antibodies by in vitro directing them onto the yeast cells and then putting the yeast into the animal is very far from reality and although it may have some biological effect, excitement for these antibodies would be so much greater if they were infused in the animal systemically either prior to after infection. Will these particular antibodies really work as potential therapeutics under this design?

(2) A second issue is the numbers of animals to read out the impact of antibodies. It is appreciated that there is attention to limitation of animals but despite some statistical differences in groups, 5 animals per group without dramatic differences in fungal burden may simply be too few of observations to be convincing in the endpoint evaluation. Also, the disease outcome score seems a little nebulous. Is it validated as a true surrogate for survival in this model? It is in fact, unlikely that this experiment was done more than one time. The investigators should defend this small number of observation or repeat the experiment again to ensure that results are robust and repeatable. It seems too much effort went into technology to have a less than robust read-out of efficacy.

Reviewer 1

“This is a well written and well documented presentation of the generation and characterization of human B cell-derived monoclonal antibodies directed against *Candida*, and I have focused my comments on the glycan microarray analyses included in this manuscript as requested by the editor.

The glycan microarray analysis was carried out by an extremely competent group with many years of experience in developing and analyzing glycan microarrays. The descriptions and documentation of the methods for printing the and analyzing the two glycan arrays mentioned are adequate and the references included cover the details of generating the targets that were printed and analyzed. That being said, I would like to make sure authors and editors are aware of an initiative, designated MIRAGE (Minimum Information Required for A Glycomics Experiment), which was created by experts in the fields of glycobiology,

glycoanalytics and glycoinformatics to produce guidelines for reporting results from the diverse types of experiments and analyses used in structural and functional studies of glycans in the scientific literature. It would be appropriate and informative for investigators reading the manuscript if the authors included a statement in this manuscript simply stating that the glycan microarray studies followed the guidelines as published by the MIRAGE initiative (Glycobiology, 2017, vol. 27, no. 4, 280–284).”

This statement has now been included in the main text (lines 241-242). Also supplementary glycan microarray document based on MIRAGE guidelines is included as Appendix in Supplementary Information.

“The microarray data were used to demonstrate the specificity of the antibodies for *C. albicans* mannoprotein and the absence of binding to other fungal or bacterial glycans presented on the array. The “validation” of the arrays were done using Dectin-1 for the Fungal, and Bacterial Polysaccharide Array (demonstrating the presence of beta1,3-glucans) and antibody PGT128 for the N-Glycan Array Set 3 (demonstrating the presence of the Man8 and Man9 N-glycans). This group would certainly have used other well-defined glycan binding proteins to demonstrate that the “probes” were printed and available for binding; these additional validation indicators should be mentioned in the supplemental information.”

The quality control of the arrays is now detailed in the supplementary glycan microarray document (Supplementary Tables 3&4).

“It would be useful to mention something regarding the statistical analyses of the results and the number of replicates of each “probe” on the arrays. This is not as critical since there were only a few positive results with large values relative to background.”

This information is provided in the legends of Supplementary Tables 3 (c) and 4 (b) which is related to the fluorescence intensities.

“It was not clear how representative mAbs were defined (page 9, line 196). There were 17 mAbs developed and 5 of them appeared directed against the Hyr1 recombinant protein and the remaining 12 bound *C. albicans* whole cells. AB121 was the only Hyr1-binding antibody tested on the two arrays, and only 7 of the other 12 that were candidate for anti-glycan binding activity were tested. The rationale for the selection could be included.”

The rationale behind selection is explained in the text (lines 237-241). Antibodies were grouped based on their VH CDR3 amino acid sequences which is the domain that is most critical for antibody binding to its target epitope. One representative from each of the 7 groups (clusters) from the total of 12 anti-whole cell mAbs were selected for further analysis where results would be indicative of the response of any of the mAbs from the corresponding group (cluster). Although the 4 Hyr1 mAbs represent unique CDR3 amino acid sequences and are therefore split into 4 different ‘clusters’, only one mAb was selected

from these 4 as we know definitively what the target antigen is for these mAbs. In terms of binding to other *Candida* species, if Hyr1 is expressed, then any of the mAbs would have bound. However, this protein is unique to *C. albicans*, and this target was selected on that basis. The targets for the anti-whole cell mAbs have yet to be fully elucidated so more focus and analysis of the different clusters was required here compared to the anti-Hyr1 mAbs.

“Finally, it interesting that AB135 binding to *C. albicans* was reduced by zymolase suggesting this antibody might be specific for a beta1,3-glucan, but beta1,3-glucans on the array were not bound by this antibody (Fig. 4a). The authors should comment on this.”

Although we observe that Zymolyase® treatment of whole cells of *Candida* resulted in some reduction in AB135 binding, we are confident that this antibody does not recognize β -1,3-glucan. We have robust probes (polysaccharides and glycans) in our microarrays that are strongly bound by the β -1,3-glucan-binding CTL Dectin-1, but not the AB135 antibody. Therefore we have no direct evidence that the antibodies recognise β -1,3-glucan. The commercial Zymolyase® we used is produced by a submerged culture of *Arthrobacter luteus* (1), and has strong lytic activity against living yeast cell walls (2,3) and results in the production of protoplasts or spheroplasts of various species. We also performed a binding experiment using a semi-purified *Candida* mannan preparation, and again we see partial reduction in AB135 binding. Therefore it is also formally possible that the enzyme preparation contains an unknown contaminant that partially breaks down the *N*-mannan.

β -1,3-glucan is the primary scaffold onto which GPI-proteins are linked via β -1,6-glucan. The enzymatic disruption of the scaffold is likely to result in the release of GPI proteins and potentially the conformation of epitopes that are proximal to the target epitope. Because we hypothesise that the epitopes are likely to be proteomannans this is indeed likely to be close to the β -1,3-glucan anchoring scaffold. We suggest this is the most likely explanation as to why AB135 binding was partially reduced by zymolyase treatment.

References:

1. Kaneko, T., Kitamura, K and Yamamoto, Y.: *J. Gen. Appl. Microbiol.*, 15, 317 (1969)
2. Kitamura, K., Kaneko, T. and Yamamoto, Y.: *Arch. Biochem. Biophys.*, 145, 402 (1971)
3. Kitamura, K., Kaneko, T. and Yamamoto, Y.: *J. Gen. Appl. Microbiol.*, 18, 57 (1972)

Minor points:

“Page 24 line 606-612 - The methods section described the two designated glycan arrays as “Fungal, and Bacterial Polysaccharide Array” and the “N-Glycan Array Set 3”. These arrays are defined in Supplementary Tables 3 and 4, but the designated titles are not actually stated in the Table legends (Table 3 legend did have a version of the designation mentioned).”

This has been corrected with designated titles stated in legends of Supplementary Tables 3 and 4.

“Page 9 line 201 – The *C. albicans* mannoprotein has an ID number of ID-13, but the text should indicate which array is being referred to.”

We have modified the text on page 9, line 244 to indicate that we are referring to ID-13 in the *N*-Glycan Array Set 3.

Reviewer 2

“This paper reports on the species-specific and pan-*Candida* human recombinant mAbs displayed properties for diagnostics and generated strong opsono-phagocytic activity of macrophages *in vitro*, and were protective in a murine model of disseminated candidiasis. Fully human antibodies would represent highly valuable reagents to explore future immunotherapies targeting medical mycoses, therefore, the manuscript is important to scientists in the specific field, and report on “human antibodies that target the major human pathogen *Candida* spp and have therapeutic and diagnostic potential” are novel. While I think this paper is of interest to those active in this important field, I have some comments requiring author's clarification and some concern about both the results and the conclusions.

1. The author mentioned that human antibody encoding V genes targeting *Candida* epitopes were cloned from single B cells derived from donors who had recovered from mucosal *Candida* infections. Actually, to select potential B cell clones producing antibodies related to protection / good prognosis, is it better to select donors who had recovered from systemic candidiasis? Please explain why the mucosal infection patients were selected.”

There is a pragmatic explanation for this. Obtaining ethical permission for patients with systemic disease was more complex and was holding up the project. In the end we were required to access samples through Pfizer's internal donor programme and our options were limited to donors in this pool. All of these donors had recovered from superficial infections. Therefore we did not have the option to work with samples from other disease groups. However, the same antibody was identified from separate B cell screens demonstrating proof-of-concept for our technology in identifying protective mAbs from the pools of B cells isolated from these donors. Furthermore, in the manuscript we show that these mAbs elicit protection in a clinically relevant animal model of systemic *Candida* infection and with advances in antibody engineering, there is the potential to enhance mAb potency if required in the future.

“2. Only *Candida albicans* and *C. dubliniensis* are truly polymorphic, due to their ability to form hyphae and/or pseudohyphae. Except *Candida albicans*, is the author aware of any other non-*albicans* *Candida* species have *Hyr1* gene?”

The *HYR1* gene was identified by members of the Aberdeen Fungal Group in 1996. It is a member of the IFF family of proteins that are found in multiple *Candida* species. However, whole genome sequencing has yet to identify a true orthologue of this gene in any other *Candida* species of fungus. Even *Candida dubliniensis*, the nearest phylogenetic relative to

C. albicans with an overall 95% genome sequence identity, lacks this gene, which is absent from the corresponding syntenic region of the chromosome (Jackson et al 2009. *Genome Research*, 19, 2231-2244). There is some residual identity to the last 30 amino acids at the corresponding position of the *C. dubliniensis* genome confirming the deletion of this gene, which is known to be of an old lineage, and not the product of a recent gene duplication in *C. albicans*. The Hyr1 protein was selected as a target for antibody screening to establish the specificity of the technology we have used. The nearest orthologues to Hyr1p in other *Candida* species have amino acid similarities which are 47-72%. However, Luo et al *PLoS One* doi.org/10.1371/journal.pone.0025909 showed that an anti-Hyr1 vaccine derived from the N-terminus of the protein, was cross-protective to other *Candida* species. We have not yet tested whether there is a protective effect of this antibody to *non-albicans* species. However, under the conditions in which we measured binding of our mAbs to other *Candida* species, we did not see any cross-reactive binding to any *non-albicans* species. We infer that the epitope that our antibodies recognise may be a specific target that is part of the deleted region in other species of IFF-like proteins that have only very weak homologies to CaHyr1p.

“3. The author’s Hyr1 related mAbs are valuable for *C. albicans* identification, however, antibodies with pan-species activity may not have much translational potential as diagnostics, because laboratory diagnosis has improved with the advent of new methods for *Candida* isolation and species identification. Rapid identification of *Candida* species has become more important because of an increase in infections caused by species other than *Candida albicans*, including species innately resistant to traditional antifungal drugs. For example, *Candida auris*, an emerging fungus that can cause invasive infections, is associated with high mortality and is often resistant to multiple antifungal drugs. Early species identification is critical for the clinical effectiveness of antimicrobial treatment.”

We agree with the reviewer that species identification is critical in diagnosis and treatment of *Candida* infections and we are aware of the new technologies being developed to do this. However, lateral flow devices have the benefits of simplicity and portability in regions of the world without access to technologically advanced ID methods (mass spec, PCR, etc). In developed countries we would not anticipate that our lateral flow diagnostic test would be used in isolation but rather in combination with current and developing tests which is standard practice in the hospital setting. We have spoken to doctors in this environment who have stated that there would be great use for a rapid (15 minute) test that could be performed while species identification was being carried out. This type of test would also have great utility in lower resourced countries where the more sophisticated technologies are not available, and this approach has been proven by the market uptake of the cryptococcal antigen dipstick test (CrAg) which works in a similar format. We have non-exclusively licensed one of our antibodies to a specialised diagnostic company for testing in a range of diagnostic formats. There would also be utility in developing these antibodies into companion diagnostics whereby patients to be treated with our therapeutic antibody are identified by testing with the same antibody in a diagnostic format.

“4. The author has shown nice data of “macrophage phagocytosis of live *C. albicans* cells pre-incubated with mAbs”; it is also important to determine macrophage candidacidal activity, especially in the presence of *hyr1*-specific mAbs. Filaments of *C. albicans* are required for tissue damage and escape killing by macrophages. Filamentous forms (hyphae and/or pseudohyphae) of *Candida* species also demonstrate increased resistance to phagocytosis compared with yeast. It will be interesting to show colony forming units (CFU) after 24^h 36 hour incubation at 37C with macrophages pre-incubated with *hyr1*-specific mAbs.”

At the suggestion of this reviewer we conducted this experiment with a *Hyr1* antibody and one of the anti-whole cell wall antibodies and compared to saline and IgG1 control and observed no statistical difference in candidacidal activity. To address this we used a murine macrophage cell line (J774.1) to be consistent with the phagocytosis experiments already carried out. We saw little effect of direct killing under the conditions used *in vitro*. However, we have increased the number of *in vivo* experiments (see below) and observed prophylactic protection *in vivo* which is a stronger indicator of therapeutic potential of the mAbs.

“5. The author concluded that macrophages travelled faster and further towards *C. albicans* yeast cells when pre-incubated with an anti-whole cell mAb, did the author observe the same -macrophages travelled faster and further -when pre-incubated with *Hyr1*- mAbs? If yes, what is the mechanism?”

As expected due to the hypha-specific expression of *Hyr1p*, there was no significant difference in uptake or time to engulfment of yeast cells that were pre-incubated with anti-*Hyr1* mAbs compared to controls. In phagocytosis assays where germ tube positive cells of *C. albicans* cells were pre-incubated with anti-*Hyr1* mAbs, time to uptake was significantly faster compared to controls. However, it was not possible to accurately measure macrophage movement towards hyphal cells in these assays due to the clumping of the fungal biomass.

The mechanism for this enhanced rate of migration of macrophages towards fungal cell targets has not yet been elucidated. However, the reviewer makes an interesting point. We have been considering the possible mechanisms that stimulate macrophage dynamic movements for some time, as discussed in previous studies [e.g. Lewis *et al.* (2012). *Stage specific assessment of Candida albicans phagocytosis by macrophages identifies cell wall composition and morphogenesis as key determinants. PLoS Pathogens* 8(3), e1002578. Rudkin *et al* (2013). *Altered dynamics of Candida albicans phagocytosis by macrophages and PMNs when both phagocyte subsets are present, mBio* 4(6):e00810-13].

We have hypothesised previously that this could be a result of contact between very fine macrophage pseudopods and *Candida* that are not visible, even at the highest microscope magnifications. We know from SEM data that macrophage projections can be up to 30 μ m long. Alternatively it could be due to altered chemo attraction once antibody is bound to the *Candida* cell wall. Sorting this out is clearly beyond the level of the current study.

“6. Please further explain why the author use a three-day mouse model of disseminated candidiasis to assess the protective efficacy of mAb in vivo. I understand it’s a new novel of invasive *C. albicans* infection of mice, and changes associated with disease become measurable within 3 days of challenge with *C. albicans*. However, evaluation of virulence effects solely in terms of kidney burdens and outcome scores seems a rather crude and unsophisticated approach. It will be interesting to see the differences in survival/ mean survival time. Furthermore, regard to two targeted organs in mouse candidiasis model mimic humans, fungal burden in the brain usually peaks on day 4 and then declined by day 7, whereas the kidney fungal burden continues to increase inexorably, reach peak by day 7.”

We appreciate the reviewer’s concerns. The 3-day model was developed in preference to crude survival models for early stage of proof-of-concept testing, to reduce animal suffering in accordance with current national trends for humane treatment of laboratory animals. In addition, the 3-day model has been used previously to successfully model systemic infection by *C. albicans* and weight loss and kidney fungal burdens were found to be excellent indicators of infection progression. These parameters are used routinely by commercial companies such as Evotec to measure protective effects of drugs and biologics.

We acknowledge the reviewer’s point on sample brain fungal burden however, brain infection only occurs at high inoculum levels – kidney burdens will continue to increase (to maximum of approx. $1e7/g$) when the mouse is close to death.

Taking on board the reviewer’s comments we conducted an additional animal experiment through Evotec which used a standard 7-day model for systemic *Candida* infection that is used routinely to test antifungal agents by challenging mice with a non-lethal dose of inoculum and evaluating protection through measuring kidney fungal burden and weight loss. Given that brain infection only occurs at high inoculum levels it did not seem appropriate to measure this in this experiment. The result of this experiment, which again showed protection and inhibition of tissue cfu accumulation, is presented in Figure 10b.

“7. Please clarify what is challenge dose for each mouse in systemic disseminated candidiasis model. The Author mentioned 1×10^4 CFU/g per mouse, does it means 1×10^4 *C. albicans* cells for each mouse or the number have to be multiplied by weight of each mouse? It’s confusing. For BALB/c mice, generally when challenged with lethal dose of *C. albicans* cells (5×10^5 per mouse), in fact the controls (all moribund) have about 10 to the seventh kidney counts and all die within 5-7 days. At fungal burden of log 4-5 of control group in this manuscript, usually mice survive well for a period of time.”

We deliberately did not choose an inoculum as high as $5e5$ or $1e6$ per mouse where death occurs rapidly, to enable us to assess the potential for protection. We agree that mice would have survived for a reasonable length of time, but our protocol prevented the mice from becoming moribund by 5-7 days post-infection. Challenge dose is routinely reported as CFU/g mouse body weight to take into account any differences in body weight. However, we accept that this needed to be clarified in the text so we have amended page 28, line 765

to say “ 3.2×10^5 cells per mouse” for the original *in vivo* experiment and written the challenge dose used in the Evotec experiment in the same format for consistency.

“In addition, the differences in kidney fungal burden among control and mAb-treated group are not impressive, only 1-1.5 log differences. The protective efficacy of anti-whole cell mAbs will be more convincing if evidenced by both kidney CFU and survival during extended time (2-3 weeks). In mouse model of disseminated candidiasis, many research groups demonstrated that vaccinations could result in marked improvement in survival and significant reductions in fungal burden during otherwise rapidly fatal haematogenously disseminated candidiasis in both immunocompetent and immunocompromised mice. Of interest are the kidney fungal burden from vaccinated mice could be under 2-3 log CFU/g. Generally, mice with kidney fungal burden indicative of a fatal infection is 6-7 log CFU/g; mice with kidney fungal burdens above this level typically die from infection, whereas mice with kidney fungal burdens below 4-5 log CFU/g could survive the infection. The conclusions of the manuscript that the species-specific and pan-Candida mAbs were protective in a murine model of disseminated candidiasis is not convincing.”

We agree that a longer-term survival experiment with higher challenge inocula and repeated dosing of the mice with the mAb would be preferable, but there are significant technical difficulties and costs associated with such experiments. We do however believe that this can be achieved and we intend to conduct such an experiment in the future. For the purposes of this manuscript the goal was to demonstrate proof-of-concept in a clinically relevant model and this has been validated by the additional experiments conducted by Evotec.

Reviewer 3

“This is a very interesting manuscript which takes antibody technology to a new level in the potential management of invasive mycoses. In a set of robust and creative experiments this work uses the isolation of single class switched memory B cells isolated from donors serum-positive for anti-Candida IgG and screened them for recognition of *hyr1* cell wall protein and a whole cell wall preparations. The reactive antibody genes(s) were cloned and expressed in kidney cells to make specific recombinant anti-Candida monoclonal antibody. A pan Candida mAbs then was shown to have opsonic activity and appeared to have protective features in a murine model of disseminated candidiasis.

The very strong features of this strategy are:

- (1) the creative technology to identify and create the potential protective mAbs. Clearly, this work showed a nice technology to keep the discovery of humanized antibodies for antifungal therapy alive and well which will be essential for future therapeutics.
- (2) By targeting Candidiasis, the investigators have not only focused on a major fungal pathogen which could use both new treatment strategies but also preventive strategies. Furthermore, there is a rich and unfulfilled history of antibody treatment in Candidiasis with the development of Mycograb® which had a positive therapeutic signal in human disease until its development was stopped.”

“(3) The manuscript is clearly written and the story is easy to follow in this presentation. There are a series of methodological maneuvers but they are explained well and need to be discussed to appreciate the value of this technology. I have little to remark on the strategy and creation of the monoclonal antibodies. Well done!

The primary issues that I would like investigators to address are the following:
It would be helpful to understand why B-cells were taken from antibody positive mucosal-infected patients. It just seemed like critically formed antibodies for treatment and prevention of invasive disease would come from patients who recovered from a candidemia and/or internal invasive candida disease. Are B-cells producing antibodies during mucosal disease the same as those responding to invasive disease? Are they as potent?”

See response above to Reviewer 2 on the same point.

“The elegance of the B-cell and antibody technology (cloning and screening) seems to be somewhat dampened by more meager animal study endpoints for efficacy. There are two issues the investigators should address.
It appears that the use of these antibodies by *in vitro* directing them onto the yeast cells and then putting the yeast into the animal is very far from reality and although it may have some biological effect, excitement for these antibodies would be so much greater if they were infused in the animal systemically either prior to after infection. Will these particular antibodies really work as potential therapeutics under this design?”

We have now done this. The purpose of the original *in vivo* experiment described in the original manuscript was a proof-of-concept to demonstrate that we could replicate the effect we saw in the *in vitro* phagocytosis assays in an animal model of infection. However we now include an additional *in vivo* single dose experiment in which the antibody was injected 4 h prior to infection. We observed protection of our anti-*Candida* antibody compared to an IgG1 control antibody as evidenced by a significant reduction in kidney fungal burden at day 7. This experiment was conducted by Evotec (UK) Ltd, using a blinded study design, thus externally validating the protective effects we have observed in-house. This generates further confidence that the antibodies are protective. This study also compared the anti-*Candida* antibody to an isotype control antibody (IgG1) rather than saline/vehicle control, further confirming that the protective effects observed were not simply due to presence of unrelated IgG1 antibody. This control has not always been used in published reports claiming protective effects of therapeutic antibodies.

“A second issue is the numbers of animals to read out the impact of antibodies. It is appreciated that there is attention to limitation of animals but despite some statistical differences in groups, 5 animals per group without dramatic differences in fungal burden may simply be too few of observations to be convincing in the endpoint evaluation. Also, the disease outcome score seems a little nebulous. Is it validated as a true surrogate for survival in this model? It is in fact, unlikely that this experiment was done more than one time. The investigators should defend this small number of observation or repeat the experiment again to ensure that results are robust and repeatable. It seems too much effort went into technology to have a less than robust read-out of efficacy.”

The group sizes for the original *in vivo* experiments were based on power calculations to detect a one log (or greater) difference in kidney fungal burden, and were also based upon the research group's expertise and previous data obtained using this infection model and mouse strain. However, we did appreciate the reviewer's comments on the small group sizes and have increased this to 10 mice per group for the additional *in vivo* experiment which was conducted by Evotec.

The outcome score has been accepted in published studies from our group, where disease outcome is determined from weight loss and kidney fungal burden. However, based on this reviewer's comment that this measure is unclear we have removed this from the manuscript but added in a graph displaying weight change during disease progression in the supplementary data.

Therefore the externally validated data from the new *in vivo* experiment combined with the data from the original *in vivo* experiment provide robust evidence for the conclusion that the anti-*Candida* mAb tested exerts a protective effect in a murine model of systemic *Candida* infection.

REVIEWERS' COMMENTS:

Reviewer #2 (Remarks to the Author):

The manuscript NCOMMS-16-15066A-Z provided the important data and could be very interesting and significant for the readers and researchers of the field. The Author has carefully and thoroughly addressed my concerns and comments. The work is well done and I believe the manuscript is ready for publication in Nature Communications.

Reviewer #3 (Remarks to the Author):

Comments

This manuscript is a well described investigation into the isolation of human B-cell derived monoclonal anti-Candida antibodies that were shown to have a biological effect under several in vivo systems. Although antibody therapy for fungal infection treatment and prevention remains unproven, this study has definitely provided a foundation to move the bar closer for realization of antibody therapy.

There are no specific criticisms of present manuscript in this review.